



**Performance of AIRS ozone retrieval over the central Himalayas: Case studies of biomass**
**burning, downward ozone transport and radiative forcing using long-term observations**
Prajjwal Rawat[1,5], Manish Naja[1], Evan Fishbein[2], Pradeep K. Thapliyal[3], Rajesh Kumar[4], Piyush
Bhardwaj[4], Aditya Jaiswal[1], Sugriva N. Tiwari[5], Sethuraman Venkataramani[6], Shyam Lal[6]
[1] Aryabhatta Research Institute of Observational Sciences (ARIES), Nainital, India
[2] NASA Jet Propulsion Laboratory, Pasadena, CA 91109, USA
[3] Space Applications Centre, ISRO, Ahmedabad 380 015, India
[4] National Center for Atmospheric Research (NCAR) Boulder, CO 80307, USA
[5] DDU Gorakhpur University, Gorakhpur 273 009, India
[6] Physical Research Laboratory (PRL), Ahmedabad, 380009, India

21 **Corresponding author:** Manish Naja (manish@aries.res.in)





**Short Summary:**
Satellite based ozone observations have gained wide importance due to their global coverage.
However, satellite retrieved products are less direct and needs to be validated, particularly in
complex terrain region. Here, ozonesonde launched from a Himalayan site is utilized to assess the
Atmospheric Infrared Sounder (AIRS) ozone retrieval. AIRS is shown to overestimates ozone in
the upper troposphere and lower stratosphere but does reasonably well in the lower troposphere
and stratosphere.





**Abstract:**
Data from ozonesondes launched at ARIES Nainital (29.40º N, 79.50º E, and 1793 m elevation)
are used to evaluate the Atmospheric Infrared Sounder (AIRS) version 6 ozone profiles and total
column ozone during the period 2011-2017 over the central Himalaya. The AIRS ozone products
are analyzed in terms of retrieval sensitivity, retrieval biases/errors, and ability to retrieve the
natural variability of columnar ozone, which has not been done so far from the Himalayan region
having complex topography. For a direct comparison, averaging kernels information is used to
account for the sensitivity difference between the AIRS and ozonesonde data. We show that AIRS
can provide quality data of ozone in the lower and middle troposphere and stratosphere with
nominal underestimation (<20%). However, in the upper troposphere and lower stratosphere
(UTLS), we observe a considerable overestimation of the magnitude as high as 102%. The
weighted statistical error analysis of AIRS ozone shows higher positive bias, root mean squared
error, and standard deviation in the upper troposphere of about 65%, 65%, and 25%, respectively.
Similar to AIRS, Infrared Atmospheric Sounding Interferometer (IASI) and Cross-track Infrared
Sounder (CrIS) are also able to produce ozone peaks and gradients successfully. However, the
statistical errors are again higher in the UTLS region that are likely related to larger variability of
ozone, lower ozone partial pressure and inadequate retrieval information on the surface
parameters. The monthly variations of columnar ozone (total, UTLS, and tropospheric) are
captured well by AIRS, except the total columnar ozone, which shows a strong bimodal variation,
unlike unimodal variation seen in ozonesonde and Ozone Monitoring Instrument (OMI). Increases
in ozone of 5 - 20% (in 2 - 6 km altitude) after the biomass burning and during events of downward
transport (in 2 - 16 km altitude) are captured well by AIRS. Ozone radiative forcing (RF) derived
from total column ozone matches well between ozonesonde (4.86 mW/m$^2$) and OMI (4.04
mW/m$^2$), while significant RF underestimation is seen in AIRS (2.96 mW/m$^2$). The fragile and
complex landscapes of the Himalayas are more sensitive to global climate change, and establishing
such biases and error analysis of space-borne sensors will help study the long-term trends and
estimate accurate radiative budgets.






## 1. Introduction:

Atmospheric ozone is an essential trace gas that plays a crucial role in the atmospheric oxidizing chemistry, air quality, and earth's radiative budget. The stratospheric ozone absorbs harmful solar ultraviolet radiation and protects biological life on earth, whereas tropospheric ozone, being a secondary air pollutant (Pierce et al., 2009) and greenhouse gas, contributes to global warming and can harm human health and crops when present in higher concentrations near the surface (Fishman et al., 1979; Ebi and McGregor 2008; Lal et al., 2017). Different radiative forcing of ozone from the stratosphere (cooling) to the troposphere (heating) (Lacis et al., 1990; Forster et al., 2007; Wang et al., 1993; Hegglin et al., 2015) demonstrate its potential importance as an atmospheric climate gas (Shindell et al., 2012). Hence, information regarding precise long-term variability in global ozone distribution is vital for better characterizing atmospheric chemistry and global climate changes (McPeters et al., 1997).

In recent decades, observations of ozone from space-borne sensors (microwave limb sounding, UV-VIS, and IR) has become an increasingly robust tool for global and higher temporal monitoring (Fishman et al., 1986; Munro et al., 1998; Foret et al., 2014). This increases our ability to analyze various influences of human activities on the atmospheric chemical composition including ozone, study their long-term impact on climate (Fishman et al., 1987), and estimate reliable radiative budgets (Hauglustaine and Brasseur 2001; Gauss et al., 2003; Aghedo et al., 2011). However, the space-based sensors are indirect and measure the atmospheric composition based upon specific algorithms utilizing radiative transfer models and a-priori information. Hence, the retrieval outputs need to be evaluated with certain reference instruments for establishing the credibility and better utilization of space-borne data.





The Himalayas, a complex terrain region, has the largest abundance of ice sheets outside Polar
regions that impacts global/regional radiative budgets and climate pervasively (e.g., Lawrence and
Lelieveld, 2010; Lelieveld et al., 2018). Here, the in-situ ground-based observations are very
sparse and limited, and complex topography along with inadequate information on the surface
parameters make it difficult to retrieve atmospheric composition from space-borne instruments.
This is because ozone weighting function, a measure of the retrieval sensitivity and a fundamental
retrieval component, depends upon various atmospheric parameters like surface temperature,
surface emissivity, and terrain height (Rodgers et al., 1976, 1990; Bai et al., 2014), which is not
uniform over the foot-print size of the AIRS (~ 13 km x 13 km) over the Himalayas. Usually, the
ozone weighting function has a shorter integrating path over the elevated terrain regions, which
follows a smaller weighting function and provides lesser sensitivity and higher errors in the final
retrievals (Coheur et al., 2005; Bai et al., 2014). Apart from the terrain height, retrieval also
depends on other factors like surface emissivity, atmospheric input constituents, input error
minimizing parameters, etc., whose accuracy matters, alters the retrieval processes abruptly, and
introduces error in the final retrieval.

The Atmospheric Infrared Sounder (AIRS) onboard the Aqua satellite has been providing reliable
vertical profiles of ozone, temperature, water vapor, and other trace gases globally twice a day
since 2002. Numerous validation studies of AIRS retrieved ozone have been carried out for
different versions since it started operating (2002). For example, Bian et al. (2007) studied AIRS
version 4 over Beijing and discussed the potential agreements (within 10%) between AIRS and
ozonesonde (GPSO3) ozone, particularly in the upper troposphere and lower stratosphere (UTLS)
region with the capability of AIRS to identify various Stratosphere-Troposphere Exchange (STE)



and transient convective events. Similarly, a study over Boulder and Lauder by Monahan et al.
(2007) using a similar AIRS version showed despite the larger biases in the lower and middle
tropospheric region, the retrieval algorithm captures the ozone variability very effectively with a
positive correlation of more than 70%. However, that study suggested a need for tropopause-
adjusted coordinates in the a-priori profiles. Both these studies (Bian et al., 2007; Monahan et al.,
2007) show larger biases of AIRS ozone in the lower and middle tropospheric regions, however,
shifts in retrieval biases and errors were seen towards the UTLS region in version 5 (Divakarla et
al., 2008), apart from significant improvements in the lower troposphere. The retrieval
methodology has also changed significantly between V4 and V5. Version 4 or earlier used
regression retrieval as the first guess in physical retrieval while later versions used a climatology-
based first guess for the physical retrieval. Also, radiative transfer models, selected channel sets,
and clarified quality indicators have been modified and improved in all successive versions.

The AIRS ozone retrieval in V5 and later has improved significantly with retrieval biases and root
mean square error (RMSE) less than 5% and 20%, respectively (Divakarla et al., 2008), over the
tropical regions. However, there is not much discussion and studies of the assessment for AIRS
ozone over the Himalayas' complex terrain, where retrieval is expected to be erroneous due to large
surface variability within its footprint. Also, most of the previous studies (Bian et al., 2006;
Divakarla et al. 2008; Pittman et al., 2009) did not utilize the averaging kernels information of the
AIRS that is vital for satellite evaluation.

Here, evaluation of AIRS version 6, which entirely depends upon the infra-red (IR) observations
after the failure of the AMSU sensor, is presented in terms of statistical analysis and ability to





retrieve the natural variability of ozone at various altitudes over the central Himalayan region using
in-situ ozonesonde observations convolved with AIRS averaging kernels. Additionally, the present
study assessed the AIRS retrieval algorithm using IASI and CrIS radiance information for one
year. AIRS columnar ozone (i.e., total, UTLS, and tropospheric columns) is also assessed with
ozonesonde, OMI, and MLS observations. AIRS has a long-term data set for ozone and
meteorological parameters, establishing such biases and error analysis is essential to make
meaningful use of its data to characterize the Himalayan atmosphere, study the trends, radiative
budgets, perform the model evaluation and data assimilation over this region.

**2 Data and Methodology**
**2.1 Data Description**
**2.1.1 AIRS**
Atmospheric Infrared Sounder (AIRS) onboard Aqua satellite, in a sun- synchronous polar orbit
at 705 km altitude, is a hyperspectral thermal infrared grating spectrometer with equatorial
crossings at ∼13:30 local time (LT). It is a nadir scanning sensor that was deployed in orbit on
May 4, 2002. AIRS along with its partner microwave instrument, the Advanced Microwave
Sounding Unit (AMSU-A), represents the most advanced atmospheric sounding system placed in
space using cutting-edge infrared and microwave technologies. These instruments together
observe the global energy cycles, water cycles, climate variations, and greenhouse gases, however
after AMSU failure the retrieval now mostly depends upon the AIRS IR observations. The AIRS
infrared spectrometer acquires 2378 spectral samples at resolutions ($\lambda/\Delta\lambda$) ranging from 1086 to
1570 cm$^{-1}$, in three bands: 3.74 µm to 4.61 µm, 6.20 µm to 8.22 µm, and 8.8 µm to 15.4 µm
(Fishbein et al., 2003; Pagano et al., 2003). The independent channels of AIRS permit retrieval of



various atmospheric states and constituents depending upon their corresponding spectral response
even in the presence of 90% cloud fraction (Susskind et al., 2003; Maddy et al., 2008). In this
study, we have used Level 2 Support physical products of AIRS (AIRS2SUP). The AIRS2SUP
files (~240 granules/day) possess extra information over the standard AIRS files, e.g., information
on averaging kernel and degree of freedom, including vertical profiles at 100 pressure levels,
against just 28 in the standard product.

The support product profiles contain 100 levels between 1100 and 0.016 mbar. While it has a
higher vertical resolution, the vertical information content is no greater than the standard product.
The information on averaging kernels and degree of freedoms (DOFs) is utilized to understand the
retrieved products more comprehensively. The DOFs of ozone, a measure of significant eigen
functions used in the AIRS retrieval, has an average value of 1.36 over the tropical latitude band
(Maddy et al., 2008) (Table S1), while over the balloon collocated region an average DOFs of 1.62
is observed (Figure S1). In the present study, best quality retrieval (O3_QC=0), associated with
cloud fraction less than 80%, and retrievals with degrees of freedom (DOF) > 0.04 are utilized.
However, analysis of cloud fraction over our collocated position shows (Figure S2) that except in
July and August the cloud fraction does not exceed ~50 ± 12%, whereas, during July and August
the maximum cloud fraction of about ~65 ± 20% is seen.









### 2.1.2 IASI (NOAA/CLASS)

The Infrared Atmospheric Sounding Interferometer (IASI) onboard MetOp satellites with a primary focus on meteorology than climate and atmospheric chemistry monitoring, is a nadir viewing Michelson interferometer (Clerbaux et al., 2007). The first MetOp satellite was launched in October 2006 (MetOp-A) and IASI was declared operational in July 2007. MetOp is a polar sun-synchronous satellite having descending and ascending nodes at 09:30 and 21:30 LT, respectively. IASI measures in the IR part of the EM spectrum at a horizontal resolution of 12 km at nadir up to 40km over a swath width of about 2,200 km. IASI covers an infra-red spectral range between 3.7 to 15.4 µm with a total of 8461 spectral channels, out of which 53 channels around 9.6 µm are utilized for ozone retrieval. IASI level 2 ozone products provided by NOAA National Environmental Satellite Data and Information Service (NESDIS) Center for Satellite Application and Research (STAR) are used in this study. The IASI (NOAA/CLASS) ozone product is retrieved based on the AIRS algorithm and has various quality control flags (Table S2). Only QC=0 data which represents a successful IR+MW retrieval is used.

### 2.1.3 CrIS/ATMS (NUCAPS)

The Cross-track Infrared Sounder (CrIS) and Advanced Technology Microwave Sounder (ATMS) sounding system onboard the Suomi NPP satellite were launched in 2011 to feature the high spectral-resolution ("hyperspectral") observations of earth's atmosphere. The CrIS instrument is an advanced Fourier transform spectrometer with an ascending node 13:30 LT and flies at a mean altitude of 824 km and performs fourteen orbits per day. It measures high-resolution IR spectra in the spectral range 650 - 2550 cm$^{-1}$ with a total of 1305 channels. The ATMS is an MW sounder with a total of 22 channels ranging from 23 to 183 GHz. These two instruments CrIS and ATMS





operate in an overlapping field-of-view (FOV) formation, with ATMS FOVs re-sampled to match
the location and size of the 3×3 CrIS FOVs for retrieval under clear to partly cloudy conditions.
Here the NUCAPS algorithm-based ozone product of CrIS is utilized. The NOAA Unique
CrIS/ATMS Processing System (NUCAPS) is a heritage algorithm developed by the STAR team
based on the AIRS retrieval algorithm (Susskind et al., 2003, 2006). The NOAA implemented
NUCAPS algorithm is a modular architecture that was specifically designed to be compatible with
multiple instruments. The same retrieval algorithms are currently used to process the AIRS/AMSU
suite (operations since 2002), the IASI/AMSU/MHS suite (operationally since 2008), and now the
CrIS/ATMS suite (approved for operations in January 2013). Here again, various quality controls
for retrieved data are provided by the NUCAPS science algorithm team, and we used QC=0
(successful IR+MW retrieval) for lesser discrepancies in our evaluation. These research products
follow a similar retrieval algorithm as developed by the AIRS science team, which gives us further
opportunity to assess the AIRS retrieval algorithm for IASI and CrIS radiances.

**2.1.4 Ozonesonde**
Electrochemical concentration cell (ECC) ozonesondes and GPS-radiosondes have been launched
from the Aryabhatta Research Institute of Observational Sciences (ARIES) (29.4º N, 79.5º E, and
1793 m elevation) Nainital (Figure 1), a high-altitude site in central Himalaya, since 2011 (Ojha
et al., 2014; Rawat et al., 2020), the only facility in the Himalayan region having regular flights.
ECC ozonesonde relies on the oxidation reaction of ozone with potassium iodide (KI) solution
(Komhyr et al., 1995) to measure ozone partial pressure in the ambient atmosphere. The typical
vertical resolution of ozonesonde is about $100 - 150$ m and has a precision of better than $\pm$ (3–5)
% with an accuracy of about $\pm$ (5–10) % up to 30 km altitude under standard operating procedures





(Smith et al., 2007). The ozonesonde is connected to iMet-radiosonde via V7 electronic interface
where radiosonde consists of GPS, PTU, and a transmitter to transmit signals to the ground. Due
to higher accuracy and in-situ measurement, ozonesonde has been widely used worldwide for
satellite and model validation (Divakarla et al., 2008; Nassar et al., 2008; Monahan et al., 2008;
Kumar et al., 2012a, 2012b; Dufour et al., 2012; Verstraeten et al., 2013; Rawat et al., 2020). Both
the ascending and descending data were recorded by ozonesonde, however, due to time lag in
descending records only ascending data is utilized (Lal et al., 2013, 2014; Ojha et al., 2014). The
data is collected at the interval of about 10 meters which is averaged over 100 meters interval using
a 3σ filter that removes the outlier values (Srivastava et al., 2015; Naja et al., 2016).

Additionally, collocated and concurrent OMI and Microwave Limb Sounder (MLS) observations
are also used to study the tropospheric ozone, UTLS, and total ozone column due to their
reasonable sensitivity and well-validated retrievals (Veefkind et al., 2006; Zeimke et al., 2006;
Fadnavis et al., 2014; Wang et al., 2021). The best quality data of MLS with data flags, i.e.,
status=even, quality>0.6, and convergence<1.18 is utilized (Barré et al., 2012). A slightly different
collocation criterion of 3°×3° grid box and daytime collocation is utilized for MLS in this work,
due to coarser resolution and to get sufficient matchups.

**2.2 Methods of Analysis**
The balloon launch time is mostly around 12:00 IST (Indian Standard Time, which is 5.5 hours
ahead of GMT). The Aqua satellite comes over the Indian region around 1:30 pm and 1:30 am
IST. Hence for collocation, only noontime (ascending) data (or ± 3 hours of balloon launch) with
1°×1° spatial collocation were chosen in this evaluation. However, for some days, there was no



noontime granule in AIRS retrieval (nearly 35 out of total 242 soundings), then we used loose
collocation of ±1 day. However, no significant changes were seen after such flexible collocation.
Most of the ozonesondes have burst altitudes near 10hPa, hence AIRS ozone profiles are evaluated
from surface to 10hPa.

Although suitable collocation criteria have been defined for a fair comparison, still different
vertical resolutions of the two data sets (ozonesonde ~100 m and AIRS ~1-5 km) make the
meaningful comparison difficult (Smit et al., 2007; Maddy and Barnet 2008). The difference in
vertical resolution and retrieval sensitivity has to be accounted for a meaningful comparison.
Hence, ozonesonde data were first interpolated at all AIRS Radiative Transfer Algorithm (RTA)
layers from surface to burst altitude, then ozonesonde profiles are smoothed according to the AIRS
averaging kernel and a-priori profile (ML climatology), leading to a vertical profile [ozonesonde
(AK)] representing what AIRS would have measured for the same ozonesonde sampled
atmospheric air mass in the absence of any other error affecting satellite observations. According
to Rodgers et al., (2000), the smoothing of the true state can be characterized as follows:
$$X_{est} = X_0 + A`(X_{sonde} − X_0) \qquad (1)$$
The AIRS provides averaging kernels information at 9 pressure levels (Figure 2b) whereas the
AIRS RTA has 100 pressure levels. So following ozone vertices (Table S3) and formulating
trapezoid matrix (Figure 2a, the details regarding the calculation of trapezoid matrices are given
in AIRS/AMSU/HSB Version 6 Level 2 Product Levels, Layers and Trapezoids), we convert 9
levels AIRS averaging kernels to 100 levels averaging kernels using following defined operation.
$$A' = F \times A_{trapezoid} \times F' \qquad (2)$$





Where $A_{trapezoid}$ and $F$ are averaging kernel matrices and trapezoid matrices ($F'$ is pseudo-inverse
of $F$). $A_{trapezoid}$ is a given product while $F$ is calculated for given ozone vertices (Table S3).

Further, in the thermal IR spectrum, the contribution of ozone or any other trace gas towards
emission/absorption of IR radiation in the radiative transfer equation depends on the exponent of
layer integrated column amounts (Maddy et al., 2008). Hence logarithmic changes in layer column
density are more linear than absolute changes. So logarithmic equations are used instead of eq. 1
for smoothing ozonesonde data in the present study.

$$\ln (X_{est}) = \ln (X_0) + A'\{\ln (X_{sonde}) - \ln (X_0)\} \qquad (3)$$

Where $X_{est}$, $X_{sonde}$, and $X_0$ are smooth ozonesonde or ozonesonde (AK), true ozonesonde, and first
guess (ML climatology) profiles, respectively.
More details on the calculation of averaging kernels can be found in AIRS documents
(AIRS/AMSU/HSB Version 6 Level 2 Product Levels, Layers and Trapezoids) or in available
literature (Maddy and Barnet 2008; Irion et al., 2008). A typical averaging kernels matrix and other
parameters are shown in Figure 2. Here Figure 2a shows a typical trapezoid matrix, Figure 2b
shows the averaging kernels at 9 pressure levels, Figure 2c shows constructed averaging kernels
at 100 RTA layers, and Figure 2d shows an example for the different ozone profiles convolved
with AKs on 15 June 2011 over the observation site.





### 2.3 Statistical Analysis

The error analysis for AIRS retrieval with interpolated and smoothed ozonesonde is based on Nalli et al. (2013, 2018). Bias, root mean squared error (RMSE), and standard deviation (STD) are studied at various RTA vertical levels from the surface to 10hPa over the Himalayan region. The finer spatio-temporal collocation utilized here has further minimized the uncertainty and error in the evaluation. Since the observation site ($29.4^{\circ}$ N, $79.5^{\circ}$ E) is at a latitude lower than $45^{\circ}$; hence there is a lesser overlap of satellite passes, and mostly a few nadir scans are close to the observation site (mostly daytime granules in range of 75 to 85). Hence all the daytime observations of AIRS are close to $\pm$ 3 hours of temporal collocation to the ozonesonde launch and possess a lesser chance of time mismatch.

Given the collocated ozone mixing ratio profiles for satellite, ozonesonde (AK) and in situ truth (ozonesonde) observations, the statistical errors are calculated as follows

$$\text{RMSE } (\Delta O_l) = \sqrt{\frac{\sum_{j=1}^{j=n} W_{l,j} \times (\Delta O_{l,j})^2}{\sum_{j=1}^{j=n} W_{l,j}}} \qquad (4)$$

$$\text{Bias } (\Delta O_l) = \frac{\sum_{j=1}^{j=n} W_{l,j} \times (\Delta O_{l,j})}{\sum_{j=1}^{j=n} W_{l,j}} \qquad (5)$$

Here $l$ runs over different RTA layers and j runs for all collocated profiles, $\Delta O_{l,j}$ the fractional deviation is taken to be the absolute deviation divided by the observed value.

$\Delta O_{l,j} = \left(\frac{O^R_{l,j} - O^T_{l,j}}{O^T_{l,j}}\right)$, where $O^T$ and $O^R$ are ozonesonde/ozonesonde (AK) and satellite retrieved ozone mixing ratio respectively.





$W_{1,j}$ is the weighting factor and assumes one of three forms $W_0 =1$, $W_1 =O^R$ and $W_2 = (O^R)^2$ and
for ozone to minimize skewing impact due to large variation in mixing ratio at different altitudes,
we have used the $W_2$ weight factor as suggested by other sounder science team (Nalli et al., 2013,

333    2018).

The Standard deviation (STD) is then calculated as follows

$$STD\ (\Delta O_l)\ =\ \sqrt{[RMSE\ (\Delta O_l)]^2 -\ [Bias\ (\Delta O_l)]^2}\qquad (6)$$

Further to check the strength of the linear relationship between the satellites retrieved data and
ozonesonde data the square of Pearson's correlation coefficient is also calculated as follows
$$r = \left[\frac{\sum_{j=0}^{j=n}\left(O_j^T - O^T avg\right)\left(O_j^R - O^R avg\right)}{\sqrt{\sum_{j=0}^{j=n}\left(O_j^T - O^T avg\right)^2\ \sum_{j=0}^{j=n}\left(O_j^R - O^R avg\right)^2}}\right]\qquad (7)$$

Where the summation is over different pairs of satellite-ozonesonde matchup values.

**2.4 Estimation of Columnar Ozone**
The total column ozone (TCO) from ozonesonde is calculated by integrating the ozone mixing
ratio from the surface to burst altitude and then adding residual ozone above burst altitude. Here
the residual ozone is obtained from satellite-derived balloon-burst climatology (BBC) (Peters et
al., 1997). The discrete integration for calculation of total ozone column (DU) between defined
boundaries is performed as follows:
$$\text{Total column ozone} = 10^7 \times \left(\frac{RT_o}{g_o P_o}\right)\times \sum_{j=1}^{j=n} 0.5\ \times (VMR[i] + VMR[i+1]) \times (P[i] - P[i+1])\ (8)$$





Where P is ambient pressure in hPa, VMR volume mixing ratio of ozone in ppbv, R (= 287.3 JKg$^{-1}$K$^{-1}$) gas constant, $g_o$ (= 9.88 ms$^{-2}$), $P_o$ (= 1.01325×10$^5$ Pa) and $T_o$ (= 273.1 K) standard temperature.
The UTLS ozone column (DU) is also calculated using Eq. (8), where the UTLS region is defined between 400 hPa to 70 hPa (Bian et al., 2007). Additionally, the tropospheric ozone column (DU) is calculated for ozonesonde utilizing the Eq. (8) with boundaries from the surface to the tropopause. The tropopause height from balloon-borne observations is estimated using the lapse rate method as well as the AIRS-derived tropopause is used and shown in Figure 3. However for OMI and MLS tropospheric ozone residual method is used which calculates the tropospheric ozone column by subtracting the OMI total column from MLS stratospheric ozone column.

362

## 3. Results and Discussion

### 3.1 Spatial Distribution: Ozonesonde and AIRS

365 The spatial distributions of ozone obtained using all ozone sounding data during four seasons are shown in Figure 4 and spatial distributions in AIRS retrieved ozone is also shown for the comparison. To obtain the AIRS ozone, the nearest swath of AIRS ozone observations is interpolated to the balloon locations and altitude. Altitude variations of the balloon along longitude is shown in Figure S3. The balloon drifts to a very long distance during winter followed by autumn and spring. During these seasons, balloon reaches to Nepal also. The wind reversal takes place during the summer-monsoon when the balloon drifted towards IGP regions (Figure 4).The spatial distributions in ozone from AIRS are more or less similar to the distributions those from ozonesonde. The bias and coefficient of determination ($r^2$) between ozonesonde and AIRS ozone is studied along the longitude and latitude (Figures S3 and S4). Lower biases (lesser than 10%)



and higher $r^2$ are seen in the lower and middle troposphere and notable biases in the upper
troposphere and lower stratosphere regions. The poor correlation (<0.4) and larger biases of up to
28% are seen at certain longitudes those are associated with higher altitudes (> 20 km). Around
the balloon launch site (Nainital, 79.45 E) highest $r^2$ score of 0.98 and low bias of 1.4% is observed,
which remain higher ($r^2$) and lower (bias) up to 80° E (Figure S3).

**3.2 Ozone Soundings and AIRS Ozone Profiles**
Figure 5 shows the average monthly ozone profiles for different collocated data sets (ozonesonde,
ozonesonde (AK), AIRS, and AIRS apriori) during seven-year periods from the surface to 10 hPa
altitude. The percentage difference between ozonesonde and AIRS ozone values at 706, 617, 496,
103, 29, and 14 hPa altitudes are mentioned and the zoomed variations in the lower tropospheric
ozone (surface to 200 hPa) are also shown in the insets. AIRS slightly (~10%) underestimate ozone
in the lower troposphere during most of the months, except the summer-monsoon (June-August),
where an overestimation of up to 20% is observed. In the middle troposphere, around 300 hPa, an
underestimation in the range 1 - 17% is seen for all months with an approaching tendency of
ozonesonde (AK) towards the true ozonesonde profiles. However, near the tropopause region,
AIRS retrievals considerably overestimate ozone by up to 102%. The overestimation was the
highest for the winter season (82 - 102%), followed by the spring, autumn, and the lowest for the
summer-monsoon season (10 - 27%). In the stratosphere, where the sensitivity of AIRS is higher
(Figure 2c), the ozonesonde and AIRS differences were relatively lower with an underestimation
in between 5 - 21%.





As expected, the difference between ozonesonde and AIRS is significantly reduced (Table 1) after
applying the averaging kernel or accounting for the sensitivity difference. This reduction was more
notable for the summer monsoon period near the tropopause, where the difference reduced from
92% to 19%, providing improvement by 72%. The improvement is as high as 100% on monthly
basis. Additionally, relative difference profiles were also analyzed for individual soundings as well
for the different seasons (Figure S5). Higher differences of about 150% between AIRS and
ozonesonde ozone observations were seen in the upper troposphere and lower stratospheric
(UTLS) region. The higher difference during winter and spring between these observations in the
UTLS region could be due to recurring ozone transport via tropopause folding over the observation
site. Such events may remain undetected by AIRS due to lower vertical resolution leading to
missing of some tropopause folding events at lower altitudes (Figure 3). However, in the lower
troposphere, larger differences between ozonesonde and AIRS during summer-monsoon are seen,
which are due to low ozone and cloudy conditions. The arrival of cleaner oceanic air during south-
west monsoon brings ozone poor air and frequent cloudy conditions over the northern India that
weakens the photochemical ozone production (Naja et al., 2014; Sarangi et al., 2014).

Figure 6 shows the yearly time series analysis of average ozone mixing ratio at three defined layers,
characterizing the middle troposphere (600 - 300 hPa), the upper troposphere (300 - 100 hPa), and
lower/middle stratosphere (100 - 10 hPa), respectively.  A prominent seasonality was seen in the
time series throughout the years, which is quite clear in the upper troposphere (300 - 100 hPa).
The ozone seasonality contrast reflects the influence of summer-monsoon and winter seasons.
Total column water vapor and monsoon index is also shown in Figure 6 and both show a tendency
of anti-correlation with ozone in 300 - 100 hPa region. The monsoon index is estimated (Wang et





al., 2001) by the difference between zonal (U) wind at 850hPa over the Arabian Sea (40E-80E,
5N-15N) and over the central Indian landmass (70E-90E, 20N-30N). The anti-correlation with
total column water vapor is slightly higher for AIRS ozone (~0.26) and it is somewhat lower with
ozonesonde (~0.15) in 300-100 hPa region. The relative difference of AIRS ozone with
ozonesonde and ozonesonde(AK) in the upper tropospheric region also shows an anti-correlation
(Figure S6) of 0.17 and 0.55 with total column water vapor and of 0.27 and 0.76 with monsoon
index, respectively.

In general, the positive values of the monsoon index correspond to strong monsoon, and negative
values correspond to weak monsoon periods (Wang et al., 2001). During the weak monsoon, there
is relatively drier air, lower cloud cover, and higher surface temperature, leading to a larger net
ozone production and relatively low upward redistributions compared to the strong monsoon
period. Thereby anti-correlation between ozone and monsoon index. The drier or low water vapor
seasons (or negative MI) show larger ozone differences between AIRS and ozonesonde, which
may be arising due to the influence of ozone-sensitive water vapor (WV) channels in mid-Infra-
red regions. Further, in the middle troposphere (600-300 hPa), a secondary ozone peak in post-
monsoon is observed, arising from the higher ozone mixing ratio after biomass burning events
over northern India (Bhardwaj et al., 2015) that seems to be missing in the AIRS ozone.

In the middle troposphere (600 - 300 hPa), AIRS retrievals show differences of more-or-less
similar magnitude with respect to both ozonesonde and ozonesonde (AK) (Figure S6). However,
in the lower/middle stratosphere (100 - 10 hPa), a considerable reduction of difference is seen after
applying the averaging kernel to ozonesonde (blue line in figure S6), which shows the need to





account for the sensitivity of AIRS in the evaluation. Further, a systematic increase of standard
deviation is also seen with the altitude. The higher standard deviations in the upper tropospheric
and stratospheric regions are mainly due to higher ozone variability associated with stratosphere-
troposphere exchange (STE) processes over the Himalayan region (Naja et al., 2016; Bhardwaj et
al., 2018).

**3.3 Statistical Analysis of AIRS Ozone Profiles**
Error analysis of AIRS retrieved ozone over the Himalayan region is performed with spatio-
temporal collocated ozonesonde observations as a reference. The methodology to calculate the
root mean square error (RMSE), bias, and standard deviation (STD) is described in section 2.3.
W2 weighting statistics is utilized due to abrupt changes of atmospheric ozone with altitude. Here
bias, RMSE, and STD between AIRS and ozonesonde are calculated at different RTA layers from
surface to 10 hPa. Figure 7 shows the average variation of bias, RMSE, and STD at different RTA
layers from surface to 10 hPa over this region. In general, higher positive bias (~65%), RMSE
(~65%), and STD (~25%) in AIRS ozone is seen in the UTLS region. In the lower and middle
troposphere, the AIRS ozone retrieval is negatively biased (0 - 25%), which increases gradually
from the surface to higher altitudes (~ 350hPa). A negative bias was also seen in the stratosphere
of about 15%. Similar to the bias, the RMSE and STD are smaller in the lower troposphere and
stratosphere with values of nearly 15% and 10%, respectively. The higher statistical errors in the
upper troposphere and the lower stratospheric region could be due to lower ozone partial pressure
and frequent stratospheric to tropospheric transport events over the Himalayas (Rawat et al., 2020),
which introduces error either after mismatch of events in AIRS coarser vertical resolution or due
to complex topography. Additionally, the AIRS tropopause frequency distribution shows less



ability of AIRS to capture deep intrusion events (Figure 3). Further, AIRS trace gas retrieval
largely depends on successful temperature retrieval and uses temperature retrieval as an input
parameter (Maddy et al., 2008). Hence, temperature retrieval error could also propagate to ozone,
and statistical error analysis of AIRS temperature shows relatively higher biases (~ 2 K) in the
upper tropospheric region (Figure S7).

The statistical error analysis was more or less similar for both true and smoothed ozonesonde
profiles. However, notable reduction and vertical shifts were also observed after applying the
averaging kernel matrix to the true ozonesonde throughout the profile. A shift of the error peak is
seen from the lower stratosphere to the upper troposphere. This could be due to the higher
sensitivity of AIRS retrieval in the lower stratosphere, which would have minimized the error at
these particular altitudes. However, in the upper troposphere, higher contribution of a-priories as
well as other factors (i.e., STE) might have resulted in larger biases and errors.

The histogram remainder between AIRS and ozonesonde is also studied at various defined layers
(Figure 8). AIRS mostly underestimates ozone with a mean bias of 2.37 ppbv, 9.29 ppbv, and
459.8 ppbv in 800 - 600 hPa, 600 - 300 hPa, and 100 - 10 hPa layers, respectively, while in the
upper troposphere (300 - 100 hPa) AIRS overestimated with a mean bias of 43.22 ppbv.
Furthermore, reminder distributions are skewed towards the negative values in the lower
stratosphere and towards positive values in the upper troposphere. More symmetric distribution
over the negative axis is observed in the middle and lower troposphere. We also studied the
correlation profiles for different seasons (Figure 8 right panel). A strong correlation is seen in the
lower and middle troposphere for spring and summer, while a poor correlation for winter and





autumn. The correlation between AIRS and ozonesonde (AK) shows a higher value in the lower
stratosphere (0.98), followed by in the upper troposphere (0.81), the middle troposphere (0.52),
and lower troposphere (0.52).

**3.4 Assessment of AIRS Retrieval Algorithm with IASI and CrIS Radiance**
The MetOp/IASI and Soumi-NPP/CrIS radiance-based ozone products are assessed using
ozonesonde data over the central Himalayan region for one year (April 2014 to April 2015),
utilizing a total of 32 soundings. Here, the IASI and CrIS based ozone retrievals are research
products provided by NOAA, whose retrieval is based on the AIRS retrieval algorithm. For IASI,
due to the 09:30 ascending nodes (morning overpass in India), ±6 h loose temporal collocation is
used. However, CrIS and AIRS follow the same collocation due to a similar noontime overpass.
The IASI, CrIS, and AIRS sensors have 8461, 1305, and 2378 IR channels respectively. Hence,
analyzing their satellite ozone products further helps to assess the AIRS retrieval algorithm for
different IR radiances and channel sets.

Figure 9a shows the seasonal ozone profiles obtained from three IR satellite sensors along with
ozonesonde for one year period. All sensors successfully captured the ozone altitude gradient and
the ozone peak height. Higher ozone concentrations during spring throughout the troposphere are
captured well by all satellite sensors. Higher ozone during spring and winter in the UTLS region
are observed by AIRS and IASI similar to ozonesonde but not by CrIS. At the same time, CrIS
sensitivity looks relatively low, where the possible role of the number of channels can be seen.
However, IASI and AIRS have effectively captured the ozone seasonal variability.



Figure 9b shows the weighted statistical error analysis of IASI, CrIS, and AIRS ozone retrieval
with the true ozonesonde observations. Here, the difference in sensitivity of the two data sets is
not accounted for as this section's primary aim is to assess the AIRS retrieved algorithm using
different IR sensor radiances and channel sets. All three space-borne sensors overestimated UTLS
ozone by more than 50%, however, in the stratosphere and lower troposphere, the bias was slightly
lower and it is somewhat underestimated. Similar to bias, the RMSE and STD were also higher in
the UTLS region by more than 80% and 60%, respectively. A consistent larger error in the UTLS
region for three IR satellite sensors that share the same radiative transfer model and retrieval
algorithm shows the possible influence of complex topography and the various STE processes in
introducing errors in retrieval processes, apart from input a-priories of the retrieval.

Additionally, Pearson correlations between ozonesonde and IASI, CrIS, and AIRS are also studied
at four atmospheric layers (i.e., 600-800 hPa, 300-600 hPa, 100-300 hPa, and 10-100 hPa) (Table
2). A relatively stronger positive correlation is found in the stratosphere (10-100 hPa), which was
highest for CrIS followed by AIRS and IASI (98%, 97%, and 91%, respectively), and a relatively
low correlation is observed in the middle troposphere (300-600 hPa) for AIRS and IASI (~ 44%
and 31%), while CrIS shows the poor correlation in the lower troposphere about 9%.

**3.5 Columnar Ozone**
**3.5.1 Total Column Ozone (TCO)**
Figure 10a shows the variations in monthly average total column ozone (TCO) from ozonesonde,
AIRS, and OMI during 2011 - 2017. In general, the TCO is higher during spring, which
subsequently drops in summer-monsoon. AIRS TCO shows a bimodal monthly variation which is





not seen in the ozonesonde and OMI observations, otherwise its monthly variation is in reasonable
agreement with ozonesonde. The OMI TCO are in good match with the ozonesonde with a
maximum difference of up to about 5 DU. Table 3 shows the difference in the TCO  between
AIRS, OMI, and ozonesonde. AIRS shows considerable overestimation in the range of 0.2 - 22
DU for some months while notable underestimation (1.8 – 10.7 DU) for others, with respect to
both ozonesonde and OMI. The correlation between AIRS TCO and ozonesonde TCO is found to
be lower (about 0.5), which has improved significantly (0.65) after applying the averaging kernel
(Table S4). To further understand the cause of bimodal variations in AIRS (higher ozone during
August, September, and October), the AIRS ozone profiles were integrated between various
altitude ranges along with corresponding ozonesonde columns, and we found the elevated total
ozone during post-monsoon is mainly contributed from the altitude above 50 hPa.

**3.5.2 UTLS Ozone Column**
Figure 10b shows the variations in monthly average UTLS ozone column for collocated and
concurrent observations of AIRS, MLS, and ozonesonde during 2011 - 2017. The UTLS region
extends between 400 hPa to 70 hPa (Bian et al., 2007). In contrast to TCO, a higher ozone in UTLS
is seen during the winter and spring (~ 45 DU) when there are recurring downward transport
events, while a clear drop of the column during the summer-monsoon shows the convective
transport of cleaner oceanic air to the higher altitudes. All the collocated observations are able to
capture the monthly variation effectively; however, there is a substantial overestimation by more
than 7 DU (Table S5) during winter and spring for both AIRS and MLS. Further, the higher
standard deviations during winter and spring show the larger variations of the ozone in the UTLS
region. Though there were notable overestimations compared to ozonesonde, still UTLS monthly





variations are captured well by AIRS with a correlation of up to 90% (Table S4). Such biases in
satellite retrieval arise due to input parameters that can be improved by using more accurate initial
parameters and surface emissivity.

**3.5.3 Tropospheric Ozone Column**
Figure 10c shows the variations in monthly average tropospheric ozone column utilizing various
collocated data sets during 2011 - 2017. The tropospheric ozone column is calculated by
integrating ozone profiles from the surface to the tropopause. WMO-defined lapse rate calculation
method is used to calculate tropopause height from balloon-borne and AIRS observations (Figure
3). Higher tropospheric ozone is observed during the spring and early summer (> 45 DU) when
annual crop-residue burning events occur over northern India, apart from downward transport from
the stratosphere. Few cases of downward transport are discussed in the next section. The
tropospheric ozone column drops rapidly during the summer-monsoon when pristine marine air
reaches Nainital. A slight increase of column is also seen during the autumn, which is again
influenced by post-monsoon crop residue burning practices in northern India (Bhardwaj et al.,
2016). The AIRS is able to capture the monthly variations very effectively; however, there are
larger biases. The biases with ozonesonde are higher when the tropopause is taken from the
balloon-borne observation, while with AIRS provides tropopause, the biases are lesser or mostly
within the one sigma limit. Like AIRS, the OMI/MLS column is in good agreement and able to
produce monthly variations; however, there are larger differences during winter and spring of more
than 10 DU. The tropospheric ozone column from ozonesonde is different for balloon-borne LRT
and AIRS tropopause, whose possible reason could be due to the lower vertical resolution of AIRS,
which will calculate tropopause with an uncertainty of 1-2 km (Divakarla et al., 2006), and on





581 average a lower tropopause pressure (or higher altitude) by 28% is calculated by AIRS compare

582 to ozonesonde measurements (Figure 3).


584 **3.6 Case Studies of Biomass Burning and Downward Transport**

585 Over the northern India, extensive agriculture practices and forest fires influence ozone at the

586 surface and higher altitudes (Kumar et al., 2011; Bhardwaj et al., 2016; Bhardwaj et al., 2018).

587 Based on MODIS fire counts, the days in between 1 March to 15 April over northern India are

588 classified as the low fire periods (LFP) as considered in previous studies over this region. The high

589 fire period (HFP) is classified when the fire counts over the observational site are more than the

590 median fire counts in the biomass burning period, typically from mid-April to mid-June (Bhardwaj

591 et al., 2016). A total of 32 and 33 collocated soundings are classified as HFP and LFP, respectively.

592 Figure 11 (left) shows the average ozone profiles up to 6 km from ozonesonde and AIRS

593 observations during HFP and LFP. The ozonesonde data show enhancement in ozone by about 5

594 ppbv to about 11 ppbv during HFP as compared to LFP that is accounting to 5-20% increase. It is

595 important to mention that enhancement is greater at higher altitude region. The enhancement is

596 slightly lower (10-15%) in AIRS profile, where most of it is contributed by the a priori profile

597 (Figure S8).

598

599 Deep stratospheric intrusion or the downward transport (DT) of ozone-rich air from the

600 stratosphere to the troposphere significantly influences ozone profiles over the subtropical regions

601 (Collins, et al., 2003; Zhu, et al., 2006). Over the subtropical Himalayas, such ozone intrusions are

602 observed during the winter and the spring seasons (Zhu et al., 2006; Ojha et al., 2014). A total of

603 10 collocated soundings are classified as DT events for ozonesonde and AIRS. Figure 11 (right)





shows ozone profiles from ozonesonde (AK) and AIRS observations for high ozone DT events as
well as the average ozone profiles of corresponding months excluding the DT event. Though there
are persistent positive biases in AIRS ozone profile compared to ozonesonde in the middle/upper
troposphere, still both the observations have captured the influence of the downward transport on
the ozone profile very effectively and show an increase in ozone of 10 - 20% in altitude range 2 -
16 km. Ozonesonde based observations have shown about two fold increase in upper-middle
tropospheric ozone due to downward ozone transport over this region (Ojha et al., 2014). Further,
the first guess profile's contribution to AIRS retrieval during DTs is negligible (Figure S8) and
shows main contribution from the observations itself. So despite the persistent biases in the AIRS
and ozonesonde observations, AIRS is able to capture the influences of downward transport (DT)
on ozone profile notably well.

**616    3.7 Ozone Radiative Forcing**

Radiative forcing is a valuable metric to estimate the radiative impacts of any anthropogenic or
natural activity on the climate system (Ramaswamy et al., 2001). It measures the net radiation at
the surface, tropopause, and the top of the atmosphere due to any atmospheric constituents. Here
we discuss the ozone radiative forcing (RF) at the surface in the ultraviolet (UV) spectral range
(Antón et al., 2013; Mateos et al., 2020) using the ozonesonde, OMI, and AIRS total column ozone
(TCO) data. The RF is calculated based on Antón et al. (2014), relative to 1979 utilizing TOMS
TOC data in 1979, monthly averaged solar zenith angles of site, clearness index based on
Chakraborty et al., (2014) and Hawas et al., (1984), and respective monthly average TCO data of
AIRS, OMI, and ozonesonde. Rather than quantifying the RF values here, our primary focus is to
show how the discrepancies of satellite ozone data (mainly AIRS) can impact the calculation of





RF values. Figure 12 shows the seasonal average ozone radiative forcing (RF) relative to 1979.
The annual average ozone RF during 2011 -2017 is 4.86, 4.04, and 2.96 mW/m$^2$ for ozonesonde,
OMI, and AIRS, respectively. The RF values for ozonesonde and OMI are comparable to Mateos
et al. (2020) (4 mW/m2) for the extratropical region. However, for AIRS, the RF value is lower
by 45%. Further, the seasonal average ozone RF (2011-2017) is consistent between ozonesonde
and OMI, while notable differences are seen in AIRS except during the winter season when
differences are marginal (Figure 12). Also from Table 3, it is clear that the higher total ozone bias
during autumn (as high as 22 DU) contributes to higher RF differences in autumn (Figure 12).

**4. Summary and Conclusions**
This study utilized 242 ozone soundings (during 2011 - 2017) conducted over the Himalayan
station (Nainital) to evaluate the AIRS version 6 ozone product and study the performance during
biomass burning events, ozone downward transport events  and estimation of ozone radiative
forcing. AIRS ozone retrieval is evaluated in terms of retrieval sensitivity, retrieval biases, retrieval
errors, and ability to retrieve the natural variability of columnar ozone at different altitude regions.
This study is first of its kind in the Himalayan region. The AIRS averaging kernels information
was applied to ozonesonde for a like-for-like comparison to overcome their sensitivity differences.
The monthly profile evaluation shows ozone peak and ozone altitude dependency is captured well
by AIRS retrieval with smaller but notable underestimation (5 - 20%) in the lower-middle
troposphere and stratosphere, while overestimation in the UTLS region as high as 102%. We show
the larger sensitivity of AIRS ozone for the summer monsoon in the UTLS region, where the biases
between AIRS and ozonesonde improved remarkably after applying AIRS averaging kernel
information.



The weighted statistical error analysis of AIRS retrieved ozone profiles shows higher positive
biases (65%), RMSE (65%), and STD (25%) in the upper troposphere. In the lower and middle
troposphere, AIRS ozone was negatively biased, apart from the stratosphere. In addition, though
the biases and errors are higher in the upper troposphere, there is a larger correlation of about 81%
showing the capability of AIRS to retrieve upper tropospheric ozone variability with certain
positive biases that can be eliminated by choosing better emissivity inputs or other retrieval inputs.
The AIRS ozone retrieval algorithm was further evaluated using the radiance of IASI and CrIS
sensors; these sensors provided similar error statistics as seen for AIRS.

The AIRS-derived columnar ozone amounts (i.e., total, UTLS, and tropospheric ozone) are also
evaluated to see whether the ozone variability at different altitude regions is being retrieved
correctly. The UTLS and tropospheric ozone monthly variations are captured well by AIRS with
certain positive biases. However, the total ozone column shows bimodal monthly variations, which
was not evident in the ozonesonde and OMI total ozone observations. Further, we show the higher
total column in AIRS during autumn, which is mostly coming from the stratospheric region above
50 hPa. The capabilities of AIRS to capture various biomass burning and downward transport
events have also been studied. AIRS captures all such events reasonably well with notable
contributions of the first guess, particularly in the biomass burning events.

Unlike the well-mixed greenhouse gases, the ozone radiative forcing (RF) remains uncertain due
to inadequate budget estimates and complex chemical processes. The total ozone discrepancies of
AIRS lead to show lower RF (by about 45%) and greater uncertainty in this Himalayan region.
Stevenson et al. (2013) have shown that a few percent uncertainties in ozone concentrations can



produce a spread of ~17% in ozone RF estimations. Here, the role of in-situ observations from
ozone soundings is shown to be important in improving the satellite retrieved ozone over the
Himalayan region by assessing and providing insights upon its error and bias. This information
could be applied for the ozone product from other satellite data set, having long-term coverage.
This will help in better understanding regional ozone and radiation budgets over this Himalayan
region having complex topography.

**Acknowledgments**
This work is supported by the ISRO-ATCTM project. Help from Deepak and Nitin in balloon
launches and coordination with the air traffic control is highly acknowledged. We are grateful to
Director ARIES for supporting this work. The National Center for Atmospheric Research is
sponsored by the National Science Foundation. SL is grateful to INSA, New Delhi for the position
and Director PRL, Ahmedabad for the support. We highly acknowledge NOAA and NASA-
EARTHDATA online data portals for providing IASI, AIRS, and CrIs label2 data. We would also
like to acknowledge the use of the MODIS fire data through FIRMS archive download. Use of
map from Google earth is also acknowledged.



**Data availability:** Satellite data are available in the respective web portal. Ozonesonde data could
be made available on a reasonable request by writing to the corresponding author.






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





**Table 1.** Ozone mixing ratio (ppbv) from ozonesonde, ozonesonde(AK) and AIRS over Nainital at six pressure levels and during winter, spring, summer-monsoon, autumn. The number of ozonesonde flights during four seasons are mentioned in the bracket.

| Pressure levels | | 706 (hPa) | 496 (hPa) | 300 (hPa) | 103 (hPa) | 29 (hPa) | 14.4 (hPa) |
|---|---|---|---|---|---|---|---|
| Winter (61) | ozonesonde | 55.1±7.3 | 54.4±5.9 | 69.5±21.4 | 238.8±116.7 | 4569.3±524.4 | 7620.6±1105 |
| | ozonesonde (AK) | 48.6±3.4 | 55.9±4.7 | 70.4±14.2 | 187.3±29.1 | 5249.1±634.1 | 8214.9±862.1 |
| | AIRS | 46.5±3.1 | 52.2±5.2 | 68.7±9.3 | 354.4±63.4 | 4428.2±456.9 | 6616.4±447.4 |
| Spring (72) | ozonesonde | 71.6±11.6 | 70.2±11.5 | 81.5±18.6 | 223.9±99.3 | 4747.0±339.6 | 8242.3±827.0 |
| | ozonesonde (AK) | 58.7±5.0 | 69.1±6.8 | 80.3±11.3 | 221.8±30.1 | 5137.8±532.7 | 8784.4±790.7 |
| | AIRS | 55.3±2.8 | 60.7±4.9 | 78.6±8.2 | 389.2±46.6 | 4687.4±314.2 | 7852.4±395.5 |
| Summer-monsoon (55) | ozonesonde | 53.0±11.8 | 65.1±15.7 | 82.1±17.4 | 138.6±24.2 | 4642.9±193.2 | 8493.6±709.4 |
| | ozonesonde (AK) | 44.1±5.7 | 62.3±9.3 | 68.7±10.6 | 224.3±13.8 | 5271.3±322.3 | 9233.8±527.8 |
| | AIRS | 48.8±1.4 | 57.5±2.1 | 63.6±2.6 | 267.4±18.9 | 4710.0±363.0 | 8333.1±577.0 |
| Autumn (54) | ozonesonde | 53.0±8.0 | 63.8±11.1 | 72.7±9.7 | 144.6±41.2 | 4439.3±195.9 | 8613.7±616.0 |
| | ozonesonde (AK) | 50.4±4.0 | 61.0±5.5 | 64.1±6.4 | 169.0±8.3 | 5086.3±242.0 | 9035.8±398.0 |
| | AIRS | 46.0±1.6 | 51.3±2.7 | 56.9±3.5 | 241.8±21.1 | 4635.4±277.0 | 7984.9±465.0 |





**Table 2.** Coefficient of determination ($r^2$) of three IR satellite sensors (AIRS, IASI and CrIS) ozone

retrieval in four broad layers with respect to ozonesonde observations.

| | Coefficient of determination ($r^2$) | | |
|---|---|---|---|
| | AIRS | IASI | CrIS |
| 600 - 800 hPa | 0.52 | 0.34 | 0.09 |
| 300 - 600 hPa | 0.44 | 0.31 | 0.22 |
| 100 - 300 hPa | 0.45 | 0.44 | 0.45 |
| 10 - 100 hPa | 0.97 | 0.91 | 0.98 |

**Table 3.** Total column ozone (TCO) differences in DU between AIRS, OMI and ozonesonde (AK),

during twelve months.

| TCO Diff. (DU) | Jan | Feb | Mar | Apr | May | Jun | Jul | Aug | Sep | Oct | Nov | Dec |
|---|---|---|---|---|---|---|---|---|---|---|---|---|
| AIRS-OMI | -3.9 | 2.2 | -1.8 | 13.2 | 16.7 | 18 | -2.2 | 17.2 | 22.1 | 13.2 | 0.0 | -2.7 |
| AIRS-ozonesonde (AK) | -6.7 | 2.2 | -2.3 | 6.5 | 17.3 | 13 | -3.1 | 20.3 | 19.1 | 10.3 | 0.2 | -10.7 |





1054

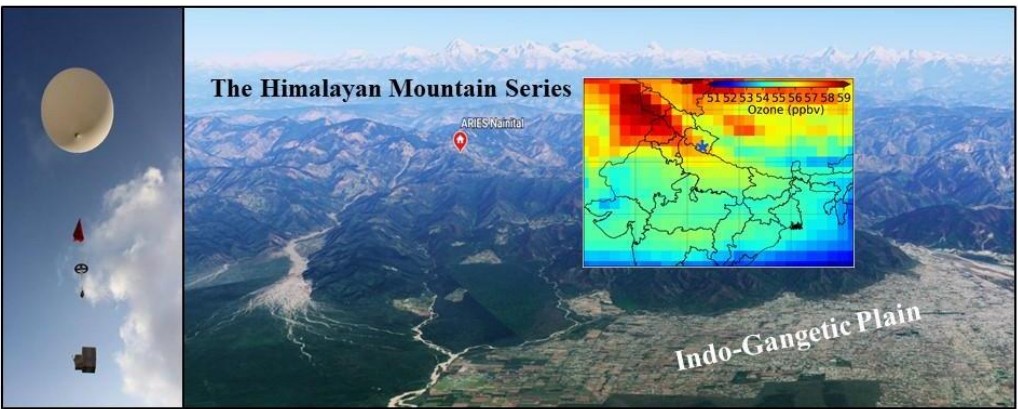

1055

**Figure 1.** Location (red color circle) of the balloon launching site (Map from © Google Earth, 2021) situated in the Aryabhatta Research Institute of Observational Sciences (ARIES) (29.4° N, 79.5° E, and 1793 m elevation), Nainital in the central Himalaya. The spatial distribution of ozone (AIRS) at 500 hPa is also shown over northern India and the location of the site is marked with a blue star. A photo of balloon, together with parachute, unwinder, ozonesonde along with GPS-radiosonde above the observation site is also shown at the left.














**Figure 2.** **(a)** Nine trapezoid functions used for ozone retrieval in AIRS-V6. **(b)** AIRS ozone averaging kernel matrix over Nainital at 9 levels vertical grid. **(c)** Calculated AIRS averaging kernel matrices at 100 RTA grids after applying the trapezoid function. **(d)** An example of ozone profiles using different data sets for 15 Jun 2011 over the observation site.



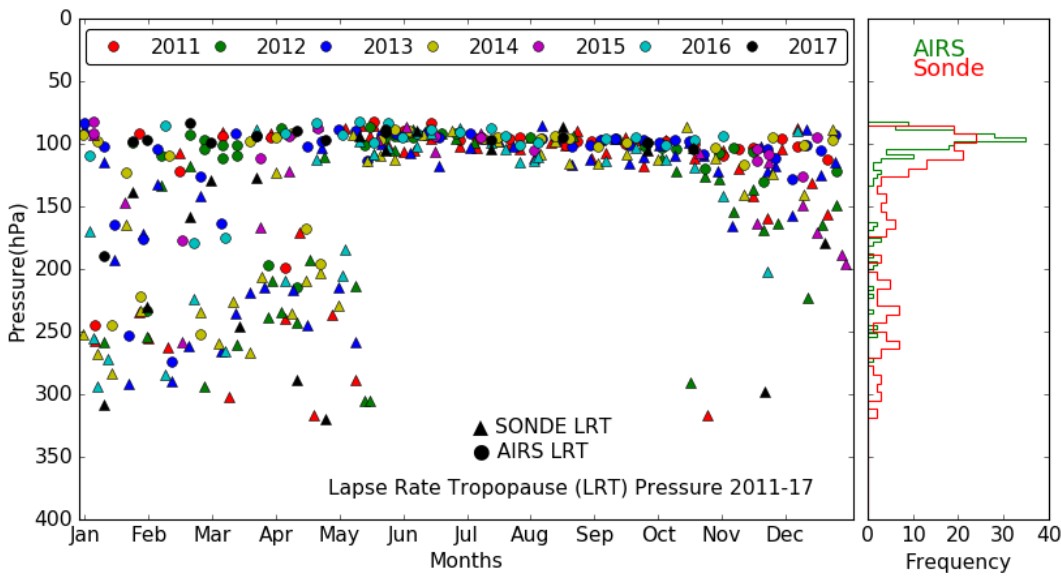


**Figure 3.** Lapse rate tropopause pressure monthly variation from balloon-borne and AIRS

observations and respective frequency distributions during 2011 - 2017.












**Figure 4.** Spatial distribution of ozone using all ozone sounding (left) launched from ARIES, Nainital, India (Map from © Google Earth, 2021). Ozone spatial distribution from AIRS (right), following the balloon track, is also shown. It could be seen that the balloon reaches Nepal many times in the winter and autumn seasons.



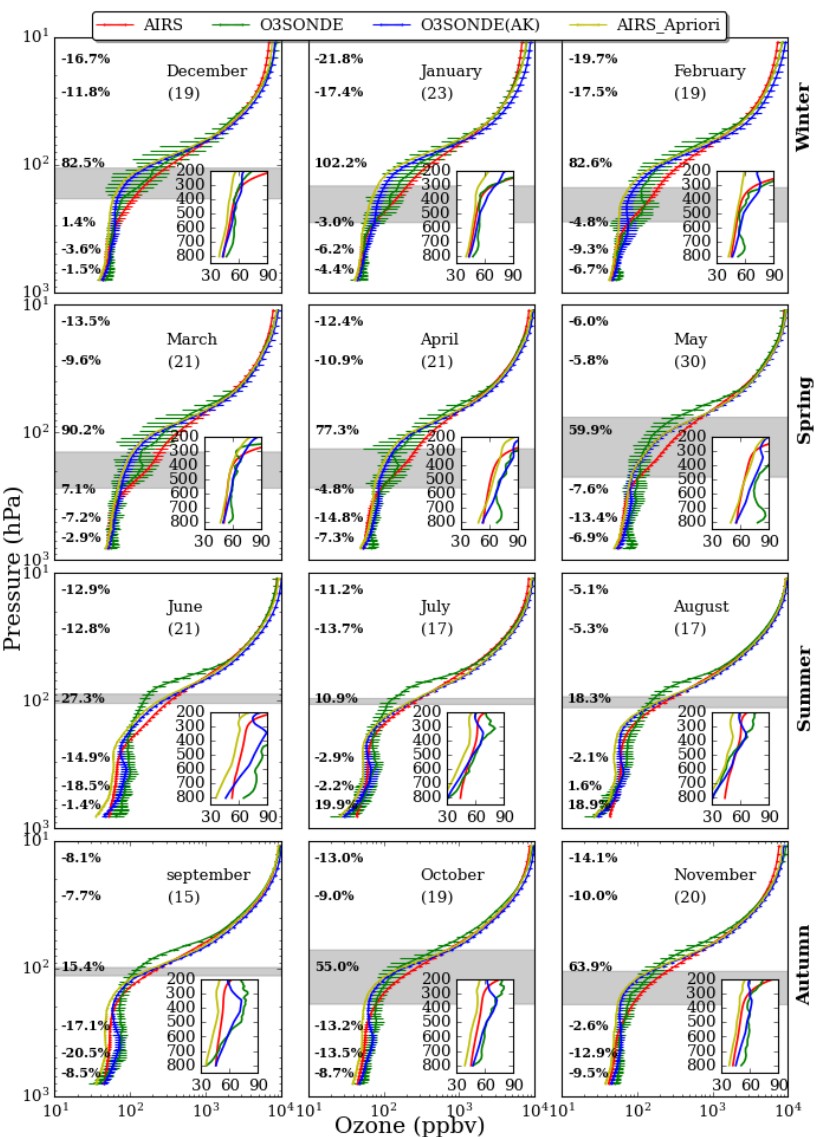

1098

**Figure 5.** Monthly averaged (2011-2017) ozone profiles of ozonesonde, AIRS, ozonesonde(AK)

and AIRS a-priori over Nainital in the central Himalaya. The percentage difference [(AIRS −

ozonesonde(AK))/ozonesonde(AK)]*100 at 706, 496, 300, 103, 29, and 14.4 hPa are also written

at respective altitudes. The number of ozonesonde for different months is written in the bracket

and grey shaded area shows the tropopause (mean±sigma) from balloon-borne observations.

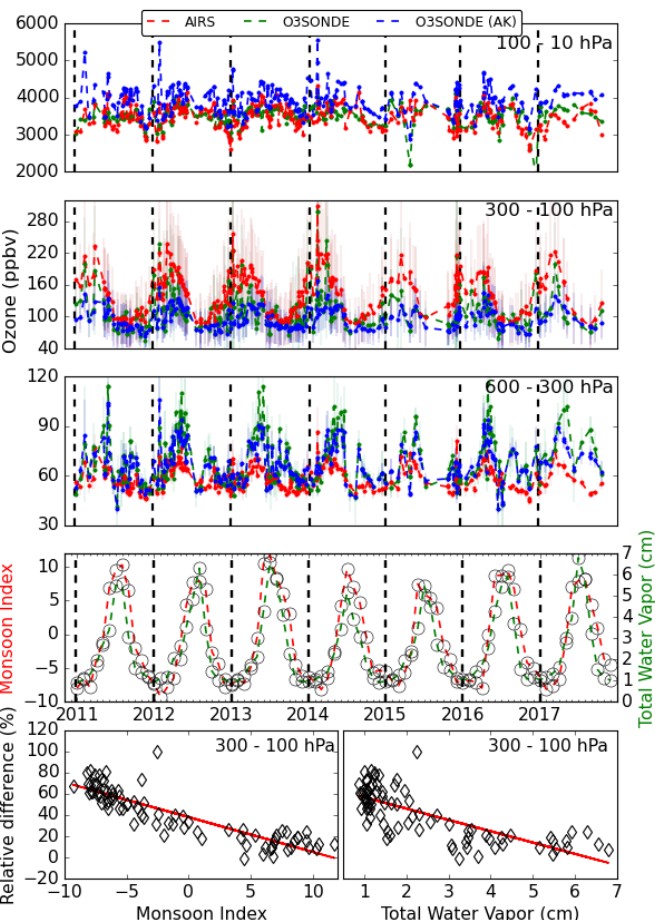

**Figure 6.** Average variations in ozone mixing ratios at three defined layers, characterizing the
lower/middle stratosphere (100 - 10 hPa), the upper troposphere (300 - 100 hPa), and the middle
troposphere (600 - 300 hPa), respectively. The monthly variation of the total column water vapor
(cm) along with the monsoon index is also shown. The scattered plot of ozone relative difference
(%) [(AIRS-O3SONDE(AK))/O3SONDE(AK)]*100, with monsoon index and total water vapor
in the upper troposphere (300 - 100 hPa) is also shown at the bottom.



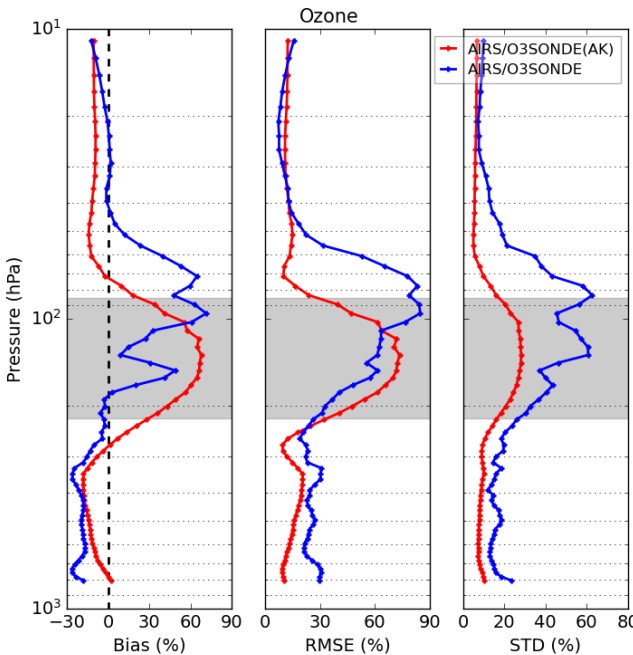


**Figure 7.** Statistical error analysis of AIRS retrieved ozone with ozonesonde and ozonesonde (AK)

for collocated data of seven years (2011 - 2017). The grey shaded area shows the tropopause region

from balloon-borne observations.





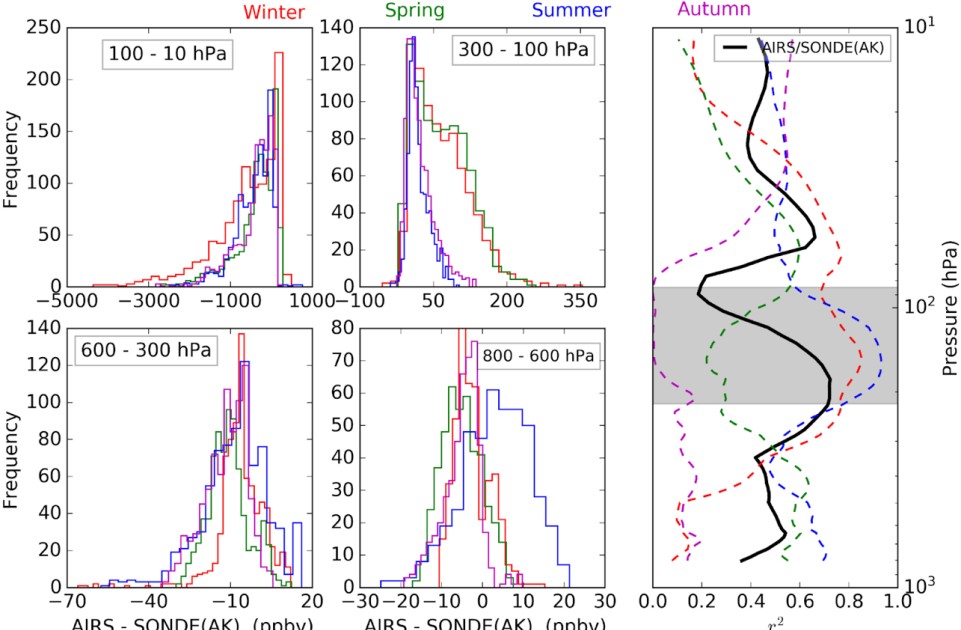

**Figure 8.** Histogram remainder between AIRS ozone and ozonesonde(AK) in the four defined

layers. The average correlation profile between AIRS ozone and ozonesonde(AK) is shown on the

right during winter (red), spring (green), summer-monsoon (blue), and autumn (magenta). The

black line is for the entire data set. The grey shaded area shows the tropopause region from balloon-

borne observations.



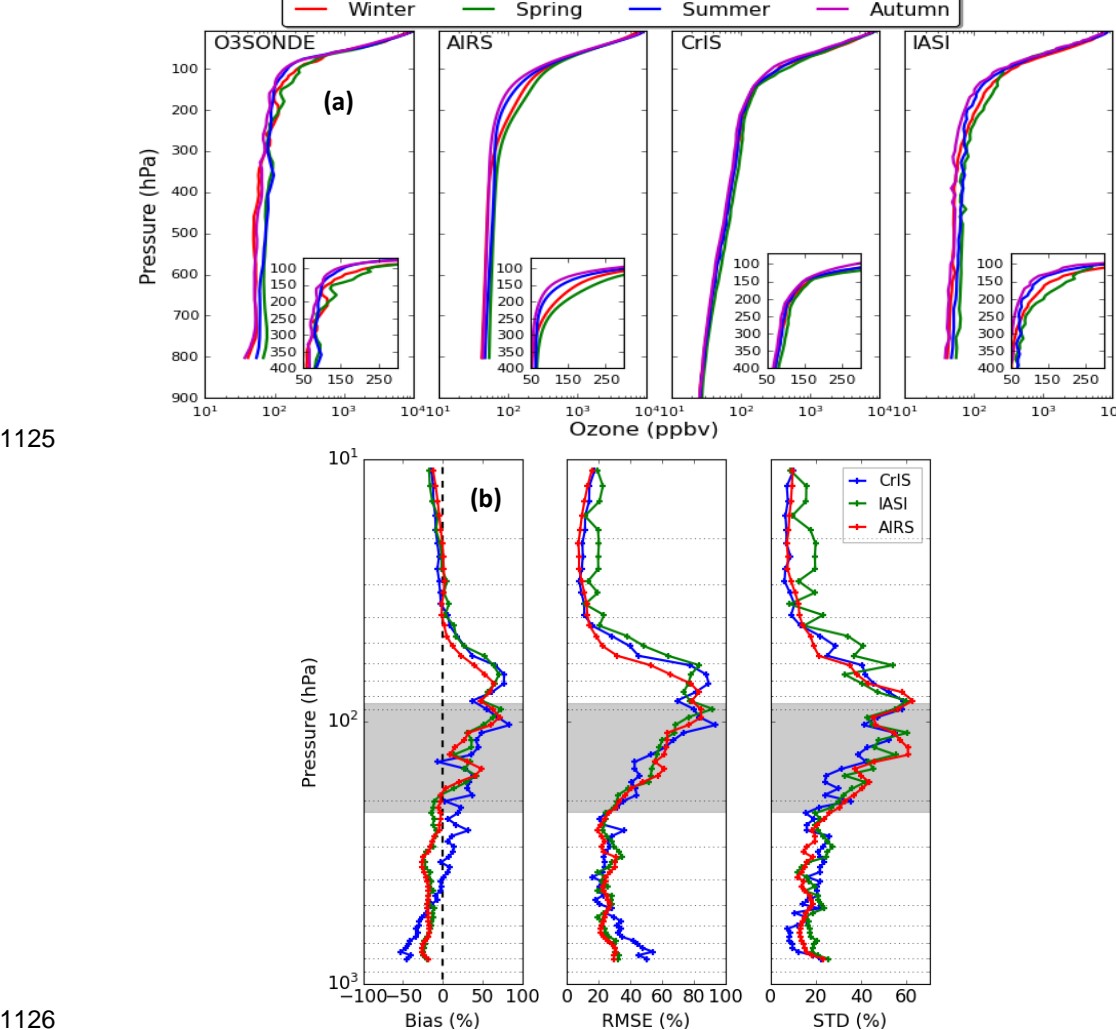

**Figure 9. (a)** Seasonal ozone profiles of three IR satellites (IASI, AIRS, and CrIS) for a smaller

sample size (April 2014 to April 2015). The IASI and CrIS products are generated using the AIRS

heritage algorithm (NOAA) and only IR+MW successful retrieval was used in quality control

(QC=0). **(b)** Statistical error analysis for the three IR satellites retrieved ozone without applying

the averaging kernel information. The grey shaded area shows the tropopause region from balloon-

borne observations.

1133

1134

1135

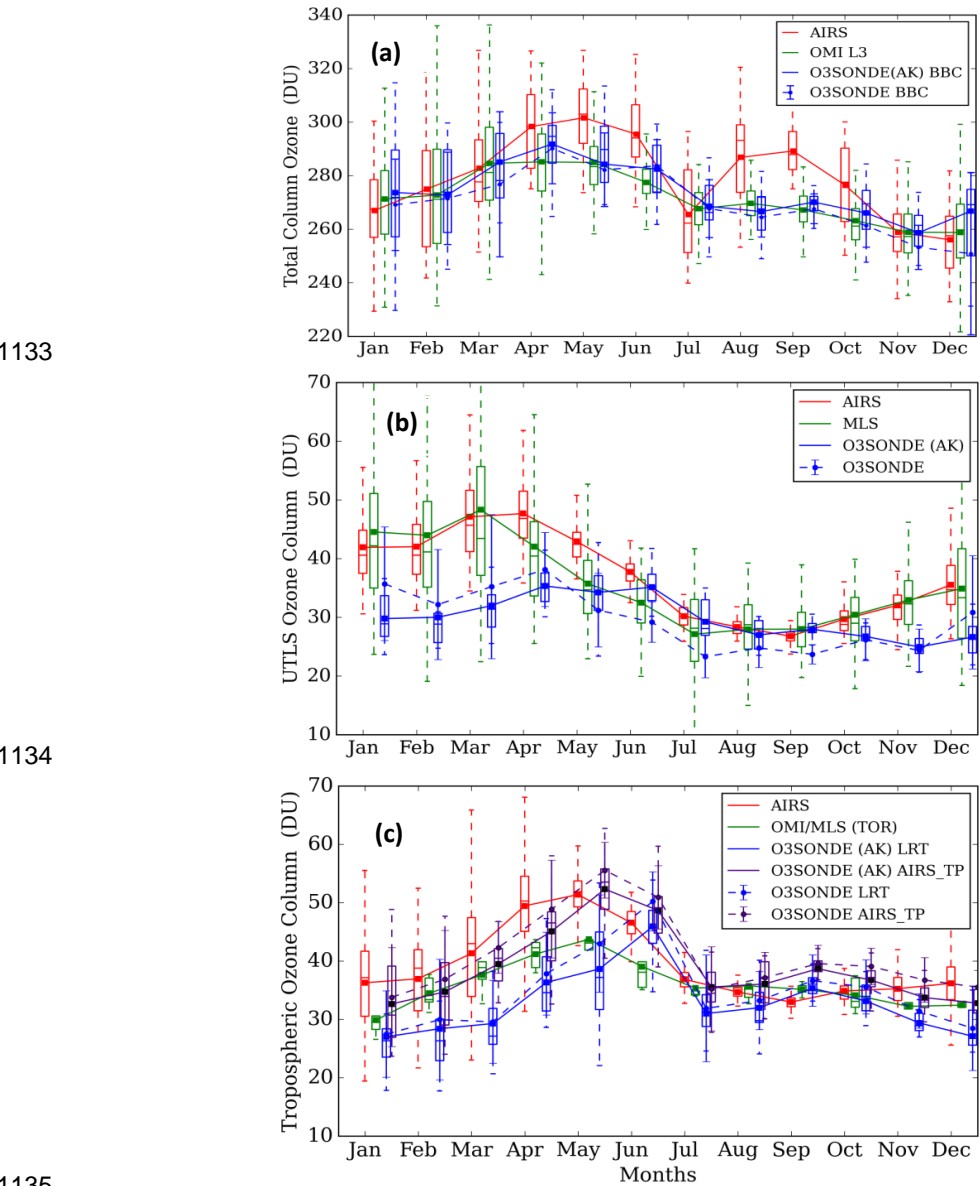

Figure 10. (a) Monthly average variation of total column ozone (TCO) for AIRS, OMI, and
ozonesonde over the central Himalaya for the seven-year period (2011-2017). (b) Monthly average
variation of UTLS ozone column for AIRS, MLS, and ozonesonde, over the central Himalayas for
the seven-year periods (2011-2017). (c) Monthly average variation of tropospheric ozone column
of AIRS, OMI/MLS, and ozonesonde, over the central Himalayas for the seven-year periods
(2011-2017). The monthly average from ozonesonde is also shown while using AIRS tropopause.




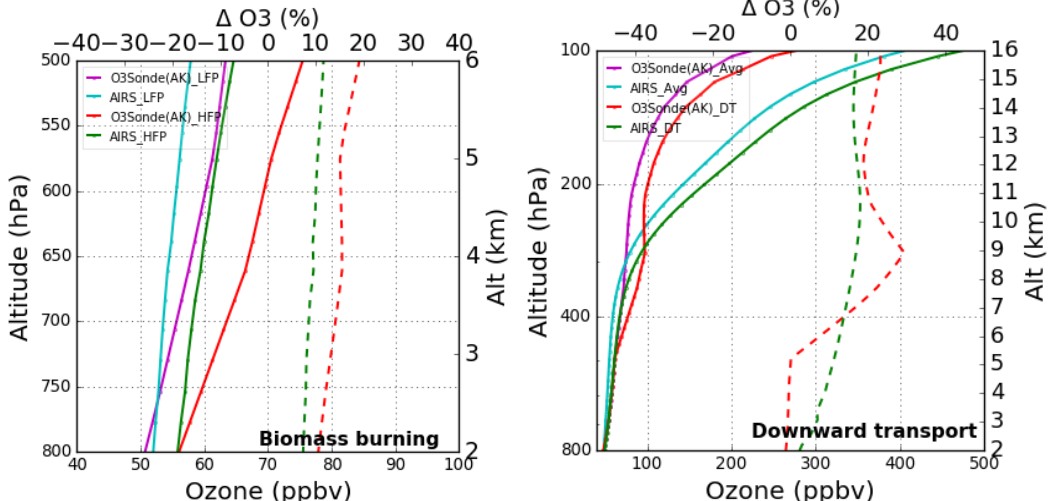


**Figure 11. (a)** Vertical ozone profiles of AIRS ozone and ozonesonde(AK) during low fire period
(LFP) and high fire period (HEP). The solid line corresponds to ozone profiles while the dotted
line shows a percentage increase in ozonesonde (red) and AIRS (green) profiles during biomass
burning events. **(b)** Vertical ozone profiles of AIRS ozone and ozonesonde(AK) during events of
downward transport. Dotted line shows ozone enhancement during downward transport events.



Page **58** of **59**






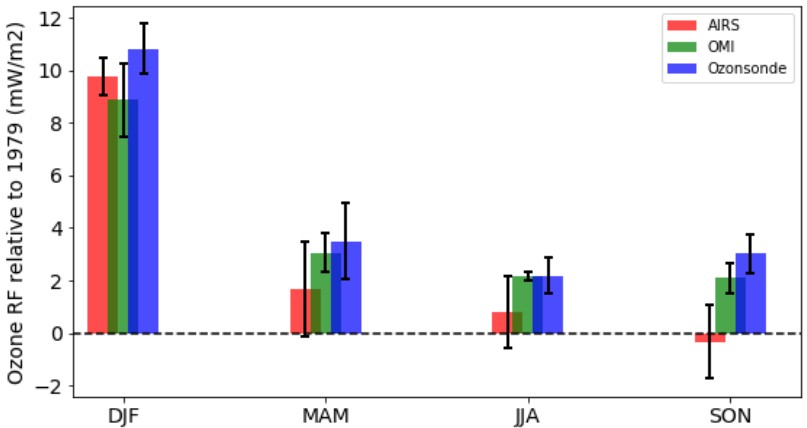

**Figure 12.** Seasonal average ozone UV radiative forcing (RF) relative to 1979 as calculated from
ozonesonde, OMI, and AIRS total ozone data during the 2011 - 2017 period.