# Peer review of "Performance of AIRS ozone retrieval over the central Himalayas: Case studies of biomass"

_Atmospheric Measurement Techniques, 2022_

## Referee Comment (RC1)

**General comments**

This paper assessed AIR ozone profile product against collocated references at the central Himalayas. They performed statistical comparisons with ozonesonde measurements and correlated satellite measurements as well as evaluated the capability of AIRS measurements to capture the atmospheric ozone variabilities inferred from summer monsoon activity, biomass burning, and stratospheric intrusions. The scope of this paper is well within AMT. However, I could not recommend this paper for publication.

**Major comments**

1. Figure 4 and section 3.1: In this section, this author discussed the spatial variation of ozone along with the ozonesonde flight path. However, it is wrong. The associated figure shows the vertical variation of ozone along with the flight path. The spots filled with green to red color represent the stratospheric air masses (o3 > 100 ppb). The horizontal drifting of balloon could be a problem in the polluted boundary layer, but the ozonesonde site used in this study is located in the Himalayan Mountain. The horizontal drifting does not matter with AIRS and ozonesonde comparison.

2. 428-437 (page 19)

   - This author related the positive values of MI with strong monsoon and negative values with weak monsoon. Actually, the monsoon index taken from Wang et al. (2001) represents the strength of the Indian summer monsoon index. The seasonal pattern of MI presented in this paper (large negative values in winter) is not consistent with that shown in Wang et al. (2001) (nearly zero in winter). You should check if there is any bug in calculating monsoon index and need a better understanding on the monsoon index of Wang et al. (2001).

   - In Figure 6, the weak summer monsoon could be associated with drier airs, but not for lower cloud cover and higher surface temperature as well as larger ozone amount near surface (larger net ozone production).

   - Line 432 "Thereby anti-correlation between ozone and monsoon index". This analysis is wrong. This anti-correlation is not driven from the interannual variations of the summer monsoon strength and its impact on ozone abundance. It is driven from the global seasonality of ozone (low in winter and high in summer) and not understandable monsoon index.

   - Line 435: Secondary ozone peak is a common feature found over the summer monsoon affected area, due to fair weather after termination of summer monsoon rainfall season and before the appearance of winter monsoon. The biomass burning could contribute on the

secondary ozone peak, but you need to demonstrate it.

3. Figure 8: I don't think that the comparison results are not inconsistent each other to characterize AIRS ozone profile quality. In manuscript, the author just describes the number of differences/R without "why", mostly.

- The AIRS-sonde differences are significantly larger at 800-600 hPa in summer than other seasons, but the correlation is larger in summer than other season. Please describe "why"

- For comparison in 300-100 hPa, the differences are much larger in spring and winter than in other season, but the correlation is significantly larger in winter and summer than in others. Please describe "why"

4. Section 3.4 Assessment of AIRS retrieval algorithm with IASI and CrIS radiance.

- line 506: Figure 9.a, the ozone peak layer is not identified.

- line 509: You should compare the averaging kernels with AIRS, IASI, and CrIS, to show the impact of different measurement characteristics on ozone profile retrievals.

- Line 523-528: In this analysis, the number of difference/R is noted, without "why".

5. Figure 10

  - This study used OMI L3 total column ozone and OMI/MLS tropospheric column ozone without any citation and acknowledge.

- This validation study should characterize the errors in AIRS total column during Fall. The bimodal peak is not found in the UTLS and troposphere. In hence, it could be inferred from stratospheric ozone retrievals. Please make a similar plot for the entire/upper/lower stratospheric column ozone and corresponding a priori column. In hence this validation study could recommend the useful vertical range of AIRS ozone profiles.

- Figure 10.b : MLS is used to evaluate AIRS column ozone integrated between 400 hPa and 70 hPa in spite that MLS is not recommended for use below 216 hPa.

- Line 560: I don't think that UTLS ozone retrievals could be improved by using more accurate surface emissivity.

- (Figure 10.c) This paper related the tropospheric ozone peak in spring and fall observed in Himalaya Mountain site with the biomass burning in northern India. I am wondering if the burning area is closed to ozonsonde site? It could be helpful to show the MODIS fire count map with ozonesonde site. In addition, please take a look at surface measurements (O3, CO) to see the seasonality caused by the biomass burning.

5. Figure 11.

- I am wondering if stratospheric intrusion cases are completely removed for comparing the ozone profiles with and without Biomass burning events (Figure 11.a) and if the burning contaminated measurements are completely removed for comparing the ozone profiles with and without downward transport events. And please specify how to define the cases of downward transport events.

6 Figure 12.

Comparing UV radiative forcing (RF) derived from OMI/AIRS/ozonesonde is meaningless in this study for evaluating the AIRS ozone profile product. That is because that Figure 10 already let us know that AIRS total ozone should be not used for scientific analysis.

**Minor comments**

1. This paper describes that the AIRS/IASI and CrIS data is based on 9.6 um, but also the applied algorithm is based on IR + MW retrievals. Please take care of this inconsistent description.

2. 187-188 (8page): It is clear to remove "associated with cloud fraction less than 80 %" in this sentence and adding "The AIRS data is flagged as best quality when cloud fraction is less than 90 % and other criteria (RMS?)".

3. 189-191 (8page): that cloud fraction does not exceed -50 +/ 12 %, except in July and Aug when cloud fraction is ~~: In manuscript, the maximum cloud fraction of ~ 65+/-20% % is highlighted. I am confused about the importance/meaning of this maximum value. The maximum value of cloud fraction could be close to 1 over the world.

4. 253-259 (11page): This paragraph is out of this 2.1.4 section ozonesonde.

5. 241-242 (10page): (3-5) % (5-10) % ➔ 3-5 %, 5-10 %

6. 382 (17 page) : different collocated data sets (~) ➔ ozonesonde and AIRS, respectively. The ozonesonde convolved with AIRS averaging kernels and AIRS a priori are also compared.

7. 385 (17 page) : Please replace "mentioned" with better one.

8. 440 (19 page) : both ozonesonde and ozonesonde (AK) ➔ ozonesondes with and without smoothing into AIRS vertical grids or original ozonesonde and smoothed ozonesondes.

9. 480(21page) : The histogram remainder between ➔ The histogram of differences between

10. 500-502 (22 page): I don't understand why the different number of entire channels between

sensors should be related to the ozone retrieval performance. All retrievals use IR near 9.6 nm.

---

## Author Comment (AC1)

**Performance of AIRS ozone retrieval over the central Himalayas: Case studies of biomass burning, downward ozone transport and radiative forcing using long-term observations**
**By Prajjwal Rawat et al., 2022 (AMTD)**

**We are grateful to both the referees for their useful comments and constructive suggestions, which have improved the MS significantly. The manuscript is suitably revised by incorporating their suggestions and comments. We are also thankful to the editors for their time. We feel that the revised manuscript is suitable for publication in AMT. Please find here our responses in boldface and the referee's comments are in regular font.**

**Refree#1**

This paper assessed AIR ozone profile product against collocated references at the central Himalayas. They performed statistical comparisons with ozonesonde measurements and correlated satellite measurements as well as evaluated the capability of AIRS measurements to capture the atmospheric ozone variabilities inferred from summer monsoon activity, biomass burning, and stratospheric intrusions. The scope of this paper is well within AMT. However, I could not recommend this paper for publication.

**We thank you for your detailed comments and suggestions on manuscripts. We have addressed all your comments and we strongly feel that our responses will be in line with your expectations.**

Major comments

1. Figure 4 and section 3.1: In this section, this author discussed the spatial variation of ozone along with the ozonesonde flight path. However, it is wrong. The associated figure shows the vertical variation of ozone along with the flight path. The spots filled with green to red color represent the stratospheric air masses (o3 > 100 ppb). The horizontal drifting of balloon could be a problem in the polluted boundary layer, but the ozonesonde site used in this study is located in the Himalayan Mountain. The horizontal drifting does not matter with AIRS and ozonesonde comparison.

**We are sorry, if there is some confusion with the terminology "spatial distribution". Here we wanted to demonstrate the overall performance of ozonesonde and AIRS over this region and felt that this is the best way to show it. As it gives feeling of spatial and vertical distributions. Our intention was not to claim this as spatial distribution alone, thereby we have clearly mentioned about the altitude in the 4th line onward in the section itself. We have also given two supplementary figures (S3 and S4) showing altitude variations, along the latitude and longitude. This section's main objective is to show ozone's spatial variation at different altitudes along the balloon track from ozonesonde and AIRS measurements. The**

ozone values are shown in the logarithmic scale from 10 ppbv to $10^4$ ppbv thereby giving a feeling of the stratospheric ozone also. Further, this figure gives both the tropospheric and stratospheric distribution along the balloon track from the two measurements. This figure also gives an overall feeling on the role of winds, its reversal and drift of the balloon during four seasons. Highly polluted IGP region is nearby and biomass burning also influences this site. Additionally, supplementary figures (S3 and S4) are the byproducts of the Figure 4 and we discuss the bias and correlations in terms of "altitude" in addition to the latitude and longitude.

To avoid any confusion, we have changed the title of the section to "Ozone Distribution along Balloon Trajectory" in the revised MS. Further, we have also revised few sentences in this section, making above aspects clearer.

2. 428-437 (page 19)

- This author related the positive values of MI with strong monsoon and negative values with weak monsoon. Actually, the monsoon index taken from Wang et al. (2001) represents the strength of the Indian summer monsoon index. The seasonal pattern of MI presented in this paper (large negative values in winter) is not consistent with that shown in Wang et al. (2001) (nearly zero in winter). You should check if there is any bug in calculating monsoon index and need a better understanding on the monsoon index of Wang et al. (2001).

The monsoon index in Wang et al. (2001) is the normalized monsoon index (MI), as mentioned in their caption of figure 3. We have confirmed the robustness of our calculated MI by comparing it with those given by Asia-Pacific Data-Research Center (APDRC) (http://apdrc.soest.hawaii.edu/projects/monsoon/daily-data.html).

APDRC MI data are based on NCAR/NCEP wind and our analysis is based on MERRA-2 reanalysis (M2TMNXSLV v5.12.4) data. In addition, we have also made calculation using ERA-5. As shown in the below figure (Figure 1), our calculated MI (by MERRA-2) are in good agreement with the MI from APDRC and also calculated with ERA-5. Small differences could arise due to the different data source (NCEP/MERRA-2/ERA-5).

Therefore, the mentioned difference is mainly due to display of "normalized monsoon index" in the figure 3 of Wang et al. (2001) and the calculated MI in the present work are correct.

[Figure]

**Figure 1. A comparison of calculated MI index in the present study (MERRA-2) with those with MI data from Asia-Pacific Data-Research Center (APDRC) and calculated using ERA-5.**

- In Figure 6, the weak summer monsoon could be associated with drier airs, but not for lower cloud cover and higher surface temperature as well as larger ozone amount near surface (larger net ozone production).

**Thank you very much. We agree that it cannot be related "directly" with the larger net ozone production and we have removed that part in the revised MS. Nevertheless, model simulations (Lu et al., 2018) have shown that the weak summer monsoon year is associated with higher surface temperature, drier air, and lower cloud cover over India. Now, we have added this (Lu et al., 2018) reference in the revised MS.**

- Line 432 "Thereby anti-correlation between ozone and monsoon index". This analysis is wrong. This anti-correlation is not driven from the interannual variations of the summer monsoon strength and its impact on ozone abundance. It is driven from the global seasonality of ozone (low in winter and high in summer) and not understandable monsoon index.

**Thank you very much for raising this concern. We would like to add a clarification here that we were not referring to the anti-correlation seen in the monthly variations. It was for annual variations. Monsoon index also refers to the total annual rainfall. Below figure 2 shows the analysis from Lu et al. (2018) in left and our analysis in right. Both show an anti-correlation between MI and the tropospheric ozone (with OMI retrieved ozone in Lu et al., 2018, the**

correlation was -0.46 over the Indian region). We also observed a significant anti-correlation between MI and annual average ozone mixing ratio in the 300 - 100 hPa region of -0.49 (Below, Figure 2 right), and a similar weaker anti-correlation is also found for other layers. Lu et al. (2018) also defined negative MI as a weaker monsoon year and positive MI as a strong monsoon year. With the help of model simulation, Lu et al. (2018) showed, as mentioned earlier, that the weak summer monsoon year is associated with higher surface temperature, drier air, lower cloud cover over India, and weaker convection, which account for higher ozone than the strong summer monsoon conditions.

Yes, it is correct that there is also a role of large scale ozone variations when showing the monthly data. We have now revised the paragraph accordingly in the revised MS.

[Figure]

**Figure 2. Annual variation of Monsoon Index and lower tropospheric ozone over India (Lu et al., 2018) on the left. Right side figure shows analysis made in the present work for 300 - 100 hPa region. The anti-correlation between ozone and monsoon index could be seen in both the analysis.**

- Line 435: Secondary ozone peak is a common feature found over the summer monsoon affected area, due to fair weather after termination of summer monsoon rainfall season and before the appearance of winter monsoon. The biomass burning could contribute on the secondary ozone peak, but you need to demonstrate it.

**Thanks again. Secondary ozone peak in the post-monsoon period has been extensively studied in surface ozone and balloon-borne observations over the present observation site. Surface ozone observations (Kumar et al., 2011), observations of its precursors like CO, NOx (Sarangi et al., 2014) and NMHCs (Sarangi et al., 2016) and model simulations (Kumar et al., 2012b) have clearly demonstrated the role of the biomass burning in this secondary peak. Balloon-borne observations have also shown the contribution of biomass burning up to about 6 km (Ojha et al., 2014). We have now briefly added this in the revised MS. Additionally, we**

have now added a supplementary figure (Figure S7) showing monthly variation in fire counts over northern India during 2011 – 2017 that clearly shows higher fire counts during pre-monsoon (spring) and post-monsoon (autumn) periods.

Sarangi, T., Naja, M., Lal, S., Venkataramani, S., Bhardwaj, P., Ojha, N., Kumar, R. and Chandola, H.C.: First observations of light non-methane hydrocarbons (C2–C5) over a high altitude site in the central Himalayas. Atmospheric Environment, 125, pp.450-460, 2016.

*(Here, we have listed additional references only those are used in the response part. References those are available in the MS are not listed here. Similar practice is followed further.)*

1. Figure 8: I don't think that the comparison results are not inconsistent each other to characterize AIRS ozone profile quality. In manuscript, the author just describes the number of differences/R without "why", mostly.

**Thanks, we have added explanation in the revised MS and we feel that the below comment is also related with this comment and we are further responding it below.**

- The AIRS-sonde differences are significantly larger at 800 - 600 hPa in summer than other seasons, but the correlation is larger in summer than other season. Please describe "why"

**Thanks for pointing this. This is possible when AIRS retrieval are highly influenced by the Apriori. We have made histogram remainder plots with AIRS retrieval and with Apriori in summer-monsoon period that do not show such difference with Apriori (below Figure 3). Additionally, the correlation coefficient is 0.86 with retrieval data (when difference is greater), while correlation is 0.65 with Apriori with negative Biases. Summer-monsoon period experiences cloudy conditions and arrival of moist/cleaner oceanic air and therefore the AIRS retrieval seems to be mostly contributed from the a-priori profile and erroneous due to cloud screening. In the revised MS we have added a sentence regarding the larger correlation of AIRS and ozonesonde (AK) during summer monsoon and possible contribution from Apriori.**

[Figure]

**Figure 3. Histogram remainder of ozonesonde(AK) with AIRS retrieved ozone and apriori in the 800 - 600 hPa region. The correlation is shown on the right during summer-monsoon.**

- For comparison in 300-100 hPa, the differences are much larger in spring and winter than in other season, but the correlation is significantly larger in winter and summer than in others. Please describe "why"

**Thank you very much. We feel that AIRS sometimes is unable to capture the prominent dynamical influence like downward transport due to its poorer vertical resolution and limited temporal resolution. Additionally, it is also observed that AIRS is unable to capture several events of the tropopause folding (Figure 3 in MS) those occurs largely in winter and early spring. The larger difference (between AIRS and ozonesonde) during winter and spring is suggested to be due to frequent dynamical events as mentioned. Additionally, such differences are not seen in the apriori as seen in the below figure 4. At the same time a higher correlation during the winter season is mainly due to better retrieval with some biases in compared to apriori. While the summer-monsoon higher correlation is mostly contributed by apriori (below Figure 4).**

[Figure]

**Figure 4. Histogram remainder of ozonesonde(AK) with AIRS retrieved ozone and apriori in the 300 - 100 hPa region. The correlation is shown on the right during winter, summer-monsoon, and spring.**

1. Section 3.4 Assessment of AIRS retrieval algorithm with IASI and CrIS radiance.

- line 506: Figure 9.a, the ozone peak layer is not identified.

**We agree that it was a general sentence and we wanted to convey that ozone peaks are broadly captured by three sensors. We have now estimated the ozone peak altitude and they are in reasonable agreement (11.35 hPa for ozonesonde, 10 hPa for AIRS, 9.11 hPa for IASI and 7.78 hPa for CrIS). Now we have added this information in the revised MS.**

- line 509: You should compare the averaging kernels with AIRS, IASI, and CrIS, to show the impact of different measurement characteristics on ozone profile retrievals.

Here, the IASI and CrIS-based ozone retrievals are research products provided by NOAA, whose retrieval is based on the AIRS retrieval algorithm. Currently, the NOAA IASI and CrIS retrieved ozone product provides no information on the averaging kernels in the level 2 product. Generally, Averaging Kernel (AK), a measure of information contents of retrieval, is calculated using multiplication between error covariance matrics and radiance jacobians, i.e., $[S_x \cdot K_n^T \cdot (K_n \cdot S_x \cdot K_n^T + S_\varepsilon)^{-1} \cdot K_n]$. Both the IASI and CrIS ozone products are based on the AIRS heritage algorithm, which utilizes the same error covariance matrices ($S_x$) for a-priories and radiance jacobians ($K_n$) in optimal retrieval; hence we believe their AKs will be more or less similar (only observational error covariance matrics ($S_\varepsilon$) will be different as it also depends upon the instruments noise equivalent differential temperature). Nalli et al. (2017) have provided the AKs information of CrIS NOAA ozone retrieval. The effective AKs of CrIS are similar to AIRS AKs, with higher sensitivity over the stratospheric region in tropical belts (Below Figure 5). Moreover, in the current MS, the differences in vertical sensitivity are not accounted for, as this section's primary aim is to assess how the AIRS retrieval algorithm performs for different IR sensor radiances and channel sets. However, a short discussion about the AKs of these data is added in section 3.4 in the revised MS.

[Figure]

Figure 5. Typical effective averaging kernels (Ae) over different regions for CrIS ozone retrieval (Nalli et al., 2017) on the left and AIRS averaging kernels over Nainital on the right.

- Line 523-528: In this analysis, the number of difference/R is noted, without "why".

**We feel that the lower correlation between ozonesonde and the satellite sensors in the lower troposphere could be due to lower sensitivity of satellite sensor and shorter lifetime of ozone. We have added this in the revised MS.**

1. Figure 10

 - This study used OMI L3 total column ozone and OMI/MLS tropospheric column ozone without any citation and acknowledge.

**Thank you very much. We have used these data from (https://acd-ext.gsfc.nasa.gov/Data_services/cloud_slice) and have cited Zeimke et al. (2006) for OMI/MLS. In the acknowledgement, we had mentioned about NASA EARTHDATA online portal for this purpose. However, now we have added a specific sentence acknowledging NASA Goddard Space Flight Center Ozone Processing Team in the revised MS.**

- This validation study should characterize the errors in AIRS total column during Fall. The bimodal peak is not found in the UTLS and troposphere. In hence, it could be inferred from stratospheric ozone retrievals. Please make a similar plot for the entire/upper/lower stratospheric column ozone and corresponding a priori column. In hence this validation study could recommend the useful vertical range of AIRS ozone profiles.

**Thank you very much for the suggestion. To study this aspect, below figure 6 shows the ozone column in four layers (100 - 70 hPa, 70 - 50 hPa, 50 - 20 hPa and 20 – 1 hPa). Bimodal peak is not seen in 100 – 70 hPa and 70 – 50 hPa layer. Two layers, above 50 hPa showed bimodal peak. In-fact, ozone peak in fall becomes more prominent in 20 – 1 hPa layer. Moreover, the AIRS apriori do not have such a bimodal peak.**

**The original MS already has this information and was mentioned that this bimodal peak is mainly due to contribution from 50 hPa and above. Nevertheless, we have further modified the sentence to make it further clearer.**

[Figure]

**Figure 6. Monthly variations of AIRS ozone column in four layers of the stratosphere.**

- Figure 10.b : MLS is used to evaluate AIRS column ozone integrated between 400 hPa and 70 hPa in spite that MLS is not recommended for use below 216 hPa.

**Thank you for pointing this out. The recommended pressure levels for scientific applications of MLS v4 ozone retrieval are 0.0215 to 261 hPa (Livesey et al., 2011; Schwartz et al., 2015). We have now revised the Figure 10b for MLS data, which is starting from 261 hPa to 70 hPa region for UTLS column.**

- Line 560: I don't think that UTLS ozone retrievals could be improved by using more accurate surface emissivity.

**Thanks. This was based on other studies (Rodgers et al., 1976, 1990; Dufour et al., 2012; Bai et al., 2014; Boynard et al., 2016, 2018) where biases in satellite retrieval are shown to be influenced by surface emissivity, apart from other factors. Dufour et al., (2012) and Boynard et al. (2018) describe that an inadequate Apriori information including surface emissivity is the most possible factor for the larger UTLS mismatch between ozonesonde and satellite data. Now we have provided these references in the revised MS.**

- (Figure 10.c) This paper related the tropospheric ozone peak in spring and fall observed in Himalaya mountain site with the biomass burning in northern India. I am wondering if the burning area is closed to ozonsonde site? It could be helpful to show the MODIS fire count map with

ozonesonde site. In addition, please take a look at surface measurements (O3, CO) to see the seasonality caused by the biomass burning.

**Long-term variations in the northern Indian biomass burning (Bhardwaj et al., 2016) and its influence on surface based observations of several trace gases (Kumar et al., 2010; Kumar et al., 2011; Kumar et al., 2012b; Sarangi et al., 2014; Sarangi et al., 2016) and aerosols (Sharma et al., 2020; Srivastava et al., 2021; Joshi et al., 2022) and balloon-borne ozone observations (Ojha et al., 2014; Bhardwaj et al., 2018) at the present observational site has been studied very extensively.**

**It has been shown that the springtime peak in fire activity over the northern Indian regions is dominated by agricultural crop residue burning and forest fires, while the secondary peak observed over the northern region during October–November is associated with crop residue burning (Kumar et al., 2011; Bhardwaj et al., 2016, 2018). The crop residue burning is a regular land clearing activity practiced in the northern Indian region following wheat and paddy crop harvesting in April–May and October–November, respectively. The spring and autumn seasons account for about 96 % of the total annual fire over the northern Indian region with 75 % in the spring season and remaining in the months of October and November (Bhardwaj et al., 2016). Furthermore, it is also demonstrated during an international field campaign (SUSKAT) that the agricultural crop residue burning in northwestern IGP led to simultaneous increases in surface ozone and CO levels at Nainital, India (present observation site) and Bode, Nepal (Bhardwaj et al., 2018). A biomass-burning-induced increase in ozone and related gases was also confirmed by model simulation and balloon-borne observations over Nainital (Kumar et al., 2011; Ojha et al., 2014; Sinha et al., 2014). In-fact, balloon-borne observations showed enhancements in ozone up to about 6 km (Ojha et al., 2014; Bhardwaj et al., 2018). These findings are also corroborated with the backward air trajectories analysis showing that the enhancement is associated with arrival of the air masses from these burning regions during the spring and autumn (e.g. Kumar et al., 2010; Sarangi et al., 2014; Bhardwaj et al., 2018).**

**Surface ozone (Kumar et al., 2010), NO, NOy, CO (Sarangi et al., 2014) and light NMHCs (Sarangi et al., 2016) showed spring and autumn peaks, though spring peak is shown to be prominent. Studies on carbonaceous aerosols also showed similar features (e.g. Dumka et al., 2015; Srivastava et al., 2021; Joshi et al., 2022). Role of biomass burning have also been shown in enhancing the regional aerosols radiative forcing (Kumar et al., 2014) and influencing the incoming solar radiation (Dumka et al., 2021).**

**Considering very extensive studies on biomass burning, with details seasonal cycle and its influence at the present observation site, we did not elaborate much in the present paper and**

**also cited limited references. However, if reviewer feel we can again add figures on MODIS fire count over the observational site.**

Kumar, R., Naja, M., Venkataramani, S. and Wild, O.: Variations in surface ozone at Nainital: A high-altitude site in the central Himalayas. Journal of Geophysical Research: Atmospheres, 115(D16), 2010.

Srivastava, P. and Naja, M.: Characteristics of carbonaceous aerosols derived from long-term high-resolution measurements at a high-altitude site in the central Himalayas: radiative forcing estimates and role of meteorology and biomass burning. *Environmental Science and Pollution Research*, *28*(12), pp.14654-14670, 2021.

Joshi, H., Naja, M., Srivastava, P., Gupta, T., Gogoi, M.M. and Suresh Babu, S.: Long-Term Trends in Black Carbon and Aerosol Optical Depth Over the Central Himalayas: Potential Causes and Implications. Frontiers in Earth Science, 10, p.851444, 2022.

Dumka, U.C., Kaskaoutis, D.G., Srivastava, M.K. and Devara, P.C.S.: Scattering and absorption properties of near-surface aerosol over Gangetic–Himalayan region: the role of boundary-layer dynamics and long-range transport. Atmospheric Chemistry and Physics, 15(3), pp.1555-1572, 2015.

Kumar, R., Barth, M.C., Pfister, G.G., Naja, M. and Brasseur, G.P.: WRF-Chem simulations of a typical pre-monsoon dust storm in northern India: influences on aerosol optical properties and radiation budget. Atmospheric Chemistry and Physics, 14(5), pp.2431-2446, 2014.

Dumka, U.C., Kosmopoulos, P.G., Ningombam, S.S. and Masoom, A.: Impact of aerosol and cloud on the solar energy potential over the central gangetic himalayan region. Remote Sensing, 13(16), p.3248, 2021.

Sharma, S.K., Choudhary, N., Srivastava, P., Naja, M., Vijayan, N., Kotnala, G. and Mandal, T.K.: Variation of carbonaceous species and trace elements in PM10 at a mountain site in the central Himalayan region of India. Journal of Atmospheric Chemistry, 77(3), pp.49-62, 2020.

1. Figure 11.

- I am wondering if stratospheric intrusion cases are completely removed for comparing the ozone profiles with and without Biomass burning events (Figure 11.a) and if the burning contaminated measurements are completely removed for comparing the ozone profiles with and without downward transport events. And please specify how to define the cases of downward transport events.

**The downward transport events mostly occur during winter (January and February) and early spring (March and early April). Ojha et al., 2016 showed a 15-year (2000–2014) analysis of an EMAC simulation to study the seasonality of ozone downward transport over the Himalayan region and showed that the frequency of downward transport is highest during the early spring pre-monsoon season.**

**In the present analysis, a total of 10 soundings are classified as downward transport (DT) events using ozonesonde observations. All these events were between January and early April. The dates for DT events are 17 Feb 2011, 01 Feb 2012, 08 Feb 2012, 13 Feb 2013, 06 Mar 2013, 15 Jan 2014, 05 Mar 2014, 06 Apr 2016, 11 Jan 2017, and 12 Apr 2017.**

Ozone soundings of 32 days (from mid-April to May) are identified as biomass burning influenced cases in the present analysis. We have now mentioned the period of DT events and biomass burning events in the revised MS.

These DT events are first classified based on an increase in the ozone vertical profile (upper-middle troposphere) and an associated drop in RH values in sonde observations. The final confirmation of DT events is made based on the MERRA-2 reanalysis data of Ertel potential vorticity (EPV), humidity, and ozone as shown in below figure 7. In general, EPV distribution is represented by the potential vorticity unit (PVU) (1 PVU = 1 × 10−6 K m2 Kg−1 s −1). Usually, air masses EPV greater than 1.6 PVU in the troposphere are suggested to be associated with the downward transport of ozone-rich air masses from the stratosphere (Cristofanelli et al., 2006). We have now briefly explained the DT criteria in the revised MS.

[Figure]

Figure 7. Ozonesonde + radiosonde ozone, RH and temperature observation on 08 Feb 2012. High ozone and low RH are observed in the vertical profile, and the MERRA-2 EPV and humidity confirm the downward transport event on the same day.

Cristofanelli, P., Bonasoni, P., Tositti, L., Bonafe, U., Calzolari, F., Evangelisti, F., Sandrini, S., and Stohl, A.: A 6-year analysis of stratospheric intrusions and their influence on ozone at Mt. Cimone (2165 m above sea level), J. Geophys. Res., 111, D03306, https://doi.org/10.1029/2005JD006553, 2006.

6 Figure 12.

Comparing UV radiative forcing (RF) derived from OMI/AIRS/ozonesonde is meaningless in this study for evaluating the AIRS ozone profile product. That is because that Figure 10 already let us know that AIRS total ozone should be not used for scientific analysis.

**Thanks. Our main purpose in this section is to demonstrate that how discrepancies in total ozone can induces the difference in the RF values. We have made this RF calculation from ozonesonde and OMI data to give feeling on RF during four seasons over this unexplored Himalayan region. We strongly feel that this section is providing useful information.**

Minor comments

1. This paper describes that the AIRS/IASI and CrIS data is based on 9.6 um, but also the applied algorithm is based on IR + MW retrievals. Please take care of this inconsistent description.

**Thanks. There are a total of 10 quality flags (i.e. 0, 1, 2, 4, 8, 9, 16, 17, 24, 25) in the NUCAPS products, where 0 represents successful infrared (IR) + microwave (MW) NUCAPS retrieval in clear sky condition, 1 represents, failed IR+MW retrieval and successful MW-only retrieval in cloudy condition, and similarly other as discussed in table S2 in the original MS. All the instruments use channels around 9.6 μm for ozone retrieval (Nalli et al., 2017). Furthermore, the AMSU (23 to 90 GHz) in MetOp and ATMS (23.8 GHz to 183.3 GHz) in NPP has no MW channels around 240 GHz, which are used for ozone retrieval in the microwave region; hence even IR+MW channel sets are used in the retrieval ozone information will only come from the 9.6 μm IR region. Nevertheless, to avoid any confusion, we have now changed IR + MW retrievals to IR retrieval in the revised MS.**

2. 187-188 (8page): It is clear to remove "associated with cloud fraction less than 80 %" in this sentence and adding "The AIRS data is flagged as best quality when cloud fraction is less than 90 % and other criteria (RMS?)".

**Thank you very much. We have now revised the sentence as suggested (section 2.1.1).**

3. 189-191 (8page): that cloud fraction does not exceed -50 +/ 12 %, except in July and Aug when cloud fraction is ~~: In manuscript, the maximum cloud fraction of ~ 65+/-20% % is highlighted. I am confused about the importance/meaning of this maximum value. The maximum value of cloud fraction could be close to 1 over the world.

**Please refer to the supplementary Figure S2 (in original MS) that shows monthly variation of the average cloud fraction over the observational site. The maximum cloud fraction of about 65 ± 20% is observed over our location. This shows that the cloud fraction crosses the 80% upper limit rarely. As mentioned above, a quality threshold set to discard the data is 80%. In the present study, only 7% of data during 2011 - 2017 has a cloud fraction of more than 80%. We have modified the sentence in the revised MS to make it clearer.**

4. 253-259 (11page): This paragraph is out of this 2.1.4 section ozonesonde.

**Thanks, we have now included section 2.1.5 as "Other Auxiliary Data" for this paragraph.**

5. 241-242 (10page): (3-5) % (5-10) % è 3-5 %, 5-10 %

**Thank, we have changed it in the revised MS.**

6. 382 (17 page) : different collocated data sets (~) è ozonesonde and AIRS, respectively. The ozonesonde convolved with AIRS averaging kernels and AIRS a priori are also compared.

**Thanks, we have changed it in the revised MS.**

7. 385 (17 page) : Please replace "mentioned" with better one.

**Thanks, we have changed it in the revised MS.**

8. 440 (19 page) : both ozonesonde and ozonesonde (AK) è ozonesondes with and without smoothing into AIRS vertical grids or original ozonesonde and smoothed ozonesondes.

**Thanks, we have now revised the sentence.**

9. 480(21page) : The histogram remainder between è The histogram of differences between

**Thanks, we have changed it in the revised MS.**

10. 500-502 (22 page): I don't understand why the different number of entire channels between sensors should be related to the ozone retrieval performance. All retrievals use IR near 9.6 nm.

**Although all the ozone retrieval is based on Spectroscopic observation of around 9.6 μm, still different satellite instruments have different resolutions for spectral observation. Instruments with a higher number of channels in the same IR region (mostly between 3.7 to 15.4 μm) have the ability to observe and detect smaller thermal contrast from different layers, depending on their weighting function. For all the instruments used in the study, the number of channels (around 9.6 μm) utilized to retrieve ozone is different, and the extra spectral information will have additive ozone information. Because of this, ultra-hyperspectral instruments are being designed for future missions. Hence, we feel that the different number of channels will influence the retrieved ozone or other retrieved parameters.**

---

## Author Comment (AC2)

**Performance of AIRS ozone retrieval over the central Himalayas: Case studies of biomass burning, downward ozone transport and radiative forcing using long-term observations**
**By Prajjwal Rawat et al., 2022 (AMTD)**

**We are grateful to both the referees for their useful comments and constructive suggestions, which have improved the MS significantly. The manuscript is suitably revised by incorporating their suggestions and comments. We are also thankful to the editors for their time. We feel that the revised manuscript is suitable for publication in AMT. Please find here our responses in boldface and the referee's comments are in regular font.**

**Refree#2**

**We thank you for your constructive comments and suggestions. The point-by-point responses to the comments are given below in boldface font.**

The authors have access to some 250 ozonesonde profiles from the central Himalayas. They were launched from a high altitude location just north of many heavily populated cities in the Indo-Gangetic Plain. Their objective is to use this valuable and unique dataset to evaluate the quality of ozone data from several satellite sensors, particularly the AIRS sensor on NASA's Aqua satellite. Though their objective is commendable, the paper suffers from several problems that include flaws in the analysis methodology, poor quality of the figures and captions, and lack of careful editing.

The authors rely very heavily on the use of the so-called "smoothing" formula (Equation 1) proposed by Rodgers and Connor (2003) published in JGR (vol 108, D3). Unfortunately, this formula is often misused. Equation 1 actually creates a hybrid of a high res profile and a priori (AP) profile. Its purpose is to asses if a remote sensing instrument has been properly calibrated and its retrieval algorithm has been correctly implemented. In such cases the retrieved and the hybrid profiles should agree. However, the formula does not provide a method of assessing the science value of the profiles independently provided by the low vertical resolution sensor. To asses it one needs to apply more traditional smoothing methods, such as Gaussian smoothing or computation of layer columns.

To understand the difference let us consider two simple examples. Let us say that a satellite sensor provides no information in a given atmospheric layer. In such cases the AK of the satellite sensor in that layer will be zero and eqn. 1 will yield the a priori (AP) value in that layer irrespective of what the ozonesonde measures. This is not what one means by "smoothing". A more relevant case is when a satellite sensor contains just the total ozone information with no useful profile information. In such cases it can be shown that eqn 1 will transform two high res profiles with very different shapes but containing the same total ozone amount to exactly the same profile that will look like the AP profile but scaled to provide the correct total ozone. Again, this is not what one

means by "smoothing".  In such cases it is best to compare total ozone values from different sensors directly.

Given this background I find only Fig 10 of the paper useful. Unfortunately, the figure is marred by several flaws. Firstly, computation of layer amounts by itself amounts to smoothing, so equation 1 should not be applied to the ozonesonde profiles. Secondly, the figure seems to show ozone variability as error bars. It is far better to plot the standard error of the mean, which is the proper method of assigning errors bars to mean values. These two changes will make the figure less cluttered and easier to evaluate.

Unfortunately, my assessment of the results presented is that the correct smoothing of the ozonesonde profiles by applying a Gaussian filter or by comparing the layer amounts (without applying eqn 1) would not confirm the key conclusion of this paper that AIRS does well in the troposphere and the stratosphere but not in the UTLS.  Still, given the uniqueness of the location, the results are worth publishing.

**Thank you very much for your elaborate comments and suggestions. In general, when comparing the measurements of two different sensors, there is no perfect way to minimize the effect of different horizontal and vertical resolutions. However, to minimize the biases due to different vertical resolutions, high-resolution profiles are generally smoothed. The AIRS IR ozone retrieval utilizes the optimal estimation-based algorithm, which have limited vertical resolution, depending upon the spectral resolution of instruments or simply the weighing function.**

**In the comparison analysis, to account for the different vertical resolutions and to perform a meaningful comparison of two independent instruments (e.g., ozonesonde and satellite), various groups have utilized the satellite averaging kernels and a-priori information (Boynard et al., 2009; Zhang et al., 2010; Verstraeten et al., 2013; Bak et al., 2019; Zhao et al., 2020) to convolved the ozonesonde or any other high-resolution instruments for smoothing their ozone profile according to Eq. 1 of MS. For example, (1) Boynard et al. (2009) utilized the averaging kernel matrix of the IASI retrievals to smooth the ozonesonde profile before their comparisons to minimize error arising from different vertical resolutions, (2)  Zhang et al., 2010, used OMI and TES AKs smoothing to compare their ozone retrieval of tropospheric ozone with ozonesonde, (3) Verstraeten et al. (2013) utilize the TES AKs to compare TES retrieved ozone profile with ozonesondes (4) Bak et al., 2019 used GEMS AKs to compare GEMS simulated tropospheric ozone profile with ozonesonde, (5) Zhao et al., 2020, utilize the TROPOMI AKs to compare TROPOMI retrieved ozone profile with ozonesondes.**

However, in some cases, very small improvements in biases are seen after applying the averaging kernels smoothing, as in the case of MLS (Adams et al., 2014), which is due to the delta functions nature of MLS averaging kernels. In such cases, the smoothing is acquired using various other techniques like the Gaussian or triangular smoothing with a full width at half maximum (FWHM) of the respective distribution equal to the typical vertical resolution of a low vertical resolution satellite instrument (Wang et al., 2020). Wang et al. (2020) for assessing SAGE III/ISS ozone retrieval with collocated satellite instruments (MLS, OMPS LP), ACE-FTS, and ozonesonde utilize the Gaussian smoothing to high-resolution profiles. While Nalli et al. (2017) utilized the broad-layer averages to compare CrIS ozone retrieval with ozonesonde.

Furthermore, we would like to mention that this is the first attempt in which ozonesondes launched over the central Himalayan site are utilized to evaluate the performance of AIRS, IASI, and CrIS ozone, particularly AIRS ozone retrieval. The AIRS averaging kernel is successfully calculated in all the 100 RTA layers using the trapezoid function and utilized for the first time in the evaluation study. There was very limited or no discussion on the AIRS ozone averaging kennels, which is a fundamental output of retrieval algorithm and possess the information of retrieval sensitivity in the previous studies (Bian et al., 2007; Monahan et al., 2007; Divakarla et al., 2008; Pittman et al., 2009).

Nevertheless, as suggested, we have applied the Gaussian smoothing to ozonesonde observations with a Gaussian distribution FWHM close to AIRS vertical resolution (~5 km, upper troposphere). Below Figure 1 shows the relative difference (RD) between AIRS ozone retrieval and smooth ozonesonde with Gaussian smoothing [O3sonde (GS)] and averaging kernel smoothing [O3sonde(AK)] for the 2011-2017 period. The RD looks smoother in the AK method than in the Gaussian method. Though the average RD profile in both the smoothing is more or less similar, the seasonal RDs are very different, which could be due to the low pass filter nature of Gaussian smoothing and Apriori contribution in AKs smoothing. In the revised MS, we have added the discussion on the choice of smoothing, and some discussion is added on the Gaussian smoothing in section 2.2.

Additionally, we agree with you that in the layer average mixing ratio and columnar ozone, the AKs smoothing must not be applied. Now smoothing is removed in all the layers and columns in the revised MS (Figure 6 and Figure 10 in the MS).

[Figure]

**Figure 1. The relative difference of AIRS and ozonesonde with Gaussian smoothing [O3sonde (GS)] and averaging kernel smoothing [O3sonde(AK)] for 2011-2017. Individual profiles are shown by a plus sign in gray color and a dashed line for the average profile for different seasons, and a thick black line for the average of all profiles.**

**Boynard, A., Clerbaux, C., Coheur, P.F., Hurtmans, D., Turquety, S., George, M., Hadji-Lazaro, J., Keim, C. and Meyer-Arnek, J.: Measurements of total and tropospheric ozone from IASI: comparison with correlative satellite, ground-based and ozonesonde observations. *Atmospheric chemistry and physics*, *9*(16), pp.6255-6271, 2009.**

**Bak, J., Baek, K.H., Kim, J.H., Liu, X., Kim, J. and Chance, K. Cross-evaluation of GEMS tropospheric ozone retrieval performance using OMI data and the use of an ozonesonde dataset over East Asia for validation. *Atmospheric Measurement Techniques*, *12*(9), pp.5201-5215, 2019.**

**Zhao, F., Liu, C., Cai, Z., Liu, X., Bak, J., Kim, J., Hu, Q., Xia, C., Zhang, C., Sun, Y. and Wang, W.: Ozone profile retrievals from TROPOMI: Implication for the variation of tropospheric ozone during the outbreak of COVID-19 in China. *Science of The Total Environment*, *764*, p.142886, 2021.**

**Adams, C., Bourassa, A.E., Sofieva, V., Froidevaux, L., McLinden, C.A., Hubert, D., Lambert, J.C., Sioris, C.E. and Degenstein, D.A.: Assessment of Odin-OSIRIS ozone measurements from 2001 to the present using MLS, GOMOS, and ozonesondes. *Atmospheric Measurement Techniques*, *7*(1), pp.49-64, 2014.**

*(Here, we have listed additional references only those are used in the response part, references those are available in the MS are not listed here. Similar practice is followed further.)*

Detailed Comments:

1) Short Summary: I have not seen any compelling evidence that AIRS does "well in the lower troposphere and stratosphere" at their site.

**Thanks. We wanted to convey it in the relative terms, when compared with the upper troposphere and the lower stratosphere. We have now revised this sentence as "AIRS is shown to overestimate ozone in the upper troposphere and lower stratosphere, while the differences with ozonesonde are lower in the middle troposphere and middle stratosphere". This statement is from the statistical analysis (MS Figure 7), where we see relatively lower biases and lower standard deviation in the middle troposphere and middle stratosphere between ozonesonde and AIRS ozone retrieval. In addition, the relative difference at broad layer average (MS Figure S6) and relative difference profile (MS Figure S5) also shows lower differences between the two measurements in the middle troposphere and middle stratosphere.**

2) Abstract: Worth mentioning the total number of sondes. These sondes, combined with sondes from other sites in India constitute a unique resource not only to evaluate satellite data but to understand the transport of ozone over north India. As I have noted above, I do not agree with the statement that "AIRS can provide quality data of ozone in the lower and middle troposphere and stratosphere" at their site. The statement "similar to AIRS, Infrared Atmospheric Sounding Interferometer (IASI) and Cross-track Infrared Sounder (CrIS) are also able to produce ozone peaks and gradients successfully" may be true at other locations, but no compelling evidence has been presented to show that it is true at their site. The statement "the monthly variations of columnar ozone (total, UTLS, and tropospheric) are captured well by AIRS, except the total columnar ozone" is confusing. It should say that monthly variation of column ozone at their site is not captured well by AIRS. The evidence that AIRS measures UTLS and tropospheric layer ozone well needs stronger justification.

**We thank you for appreciating our efforts towards continuous balloon-borne ozone soundings over the complex Himalayan terrain. Following your suggestion, the total number of sondes is mentioned in the abstract.**

**Similar to the previous response, the sentence "AIRS can provide quality data of ozone in the lower and middle troposphere and stratosphere" is revised to "AIRS has lower difference with ozonesonde ozone in the lower and middle troposphere and stratosphere with nominal underestimations of less than 20%".**

**Regarding the ozone peak and gradient, we have estimated the ozone gradient and the below figure 2 shows the vertical distribution of the running ozone gradient. The gradient profiles**

are more-or-less similar during four seasons. The estimated annual average ozone gradient in regions between tropopause to gradient peak are 231.5 ppbv/hPa, 199.0 ppbv/hPa, 193.2 ppbv/hPa and 199.1 ppbv/hPa for ozonesonde, AIRS, CrIS, and IASI, respectively. Similarly the ozone peak altitudes are 11.35 hPa, 10 hPa, 9.11 hPa, and 7.78 hPa for ozonesonde, AIRS, IASI, and CrIS, respectively. We have now added these information in the revised MS (section 3.4).

[Figure]

**Figure 2. Ozone gradient profile along the AIRS RTA pressure levels from ozonesonde, AIRS, CrIS, and IASI.**

About the monthly variations of columnar ozone, we have now revised this sentence. The revised sentence is "Furthermore, AIRS fail to capture the monthly variation of the total ozone column, with a strong bimodal variation, unlike unimodal variation seen in ozonesonde and Ozone Monitoring Instrument (OMI). In contrast, the UTLS and tropospheric ozone column are in reasonable agreement."

In addition, though there are persistence biases, particularly for the UTLS column, the correlation of UTLS (between AIRS and ozonesonde) is very strong (0.75) (below figure 3). In addition, we have performed an additional estimate for the correlation at each pressure levels in UTLS region and r2 is 0.82.

About the tropospheric column comparison, Figure 10c in the original MS shows monthly variation in ozonesonde based tropospheric column ozone using "two" tropopause (i) sonde based tropopause (ii) AIRS based tropopause. It is clear that monthly variation in tropospheric ozone column with AIRS based tropopause shows much better agreement in comparison of sonde based tropopause. The correlation (below figure 3) between AIRS and

**ozonsonde is much better (0.72) when used AIRS tropopause. We have added this information in the revised MS (section 3.5.3 and Table S4)**

**Hence we feel that the AIRS UTLS and tropospheric ozone column information are reasonably agreeing with ozonesonde. Nevertheless, we have now revised the sentence in the abstract as mentioned above.**

[Figure]

**Figure 3. UTLS ozone column correlation (left) and the tropospheric ozone column (right) between AIRS and ozonesonde.**

3) Table 1: The caption needs to indicate what is mean by the numbers following ± sign. I assume they are standard deviations, not standard error of the mean. In that case the standard error would be much smaller and even small differences would become statistically significant. As discussed above, the agreement in the lower layers does not necessarily imply that AIRS is doing a good job. It may only imply that AIRS AP is consistent with ozonesonde. Large differences near 100 hPa is a concern, since it implies some sort of problem with the AIRS retrieval algorithm.

**We again thank you. Yes, the numbers following the ± sign are standard deviations. Now as suggested, we have estimated the standard errors and used in the revised MS.**

**We agree with you and this has also been described in above few comments. We have also modified the sentences (section 3.3) and abstract accordingly.**

4) Table 2: If these values were derived after applying AK to the ozonesonde data, then it would be very useful to provide the values with and without applying AK, since the latter values are what a user of AIRS data would actually care about. It makes no sense to me to average the MR in the 10-100 hPa layer. Since the MR drops by nearly two orders of magnitude between 10 and 100 hPa, the average would essentially be the value near 10 hPa. It is much better to compare the ozone column in this layer (without applying AK).

**Thank you very much. Table 2 shows the $R^2$ values of ozonesonde with AIRS, IASI, and CrIS, respectively, without applying AKs. We have now mentioned this in the caption of table 2. As indicated, we have now divided the 100 - 10 hPa region into the two layers (lower stratosphere (100 - 50 hPa) and middle stratosphere (50 - 10 hPa)). We have also revised other figures (Figure 6, Figure 8, and Figure S6) and added this information in sections 3.2 and 3.3 ) in the revised MS.**

5) Table 3: In comparing columns one should not apply AK.

**We thank and appreciate this suggestion. We agree with the reviewer that the smoothing should not be applied in comparing columns. We have now revised the table 3 and similar changes are also done in supplementary tables (Table S4 and S5).**

6) Figure 2c: This figure very clearly shows the problem one has in interpreting AIRS ozone profile data. Since the AKs peak near the ozone density peak, the primary information contained in AIRS measurement is the column ozone amount. The profile information is extremely limited. However, if the variability of (log of) ozone near the peak is small, the secondary peak at 200 hPa may help capture some of the variability near that level. While the short-term variability of O3 near the density peak is probably quite small (this needs to be checked using sonde data), it is important to note that QBO in O3 occurs near the ozone density peak. So, the peak in the AK near the peak may introduce QBO like signals at the lower levels.

**The AIRS ozone averaging kernels are calculated and utilized for AIRS ozone evaluation over the central Himalayan region. To our knowledge, the AIRS ozone AKs at all 100 RTA layers are constructed and discussed here for the first time. Generally, Averaging Kernel (AK), a measure of information contents of retrieval, is calculated using multiplication between error covariance matrics and radiance jacobians, i.e., $[S_x \cdot K_n^T \cdot (K_n \cdot S_x \cdot K_n^T + S_\varepsilon)^{-1} \cdot K_n]$. In each AIRS profile retrievals, the error covariance matrices will be nearly same depending on apriori informations, while the radiance jacobians will be slightly different. Hence for each retrieval, a little different shape of AKs is expected, with nearly similar information contents. We agree with the reviewer that the AIRS ozone retrieval is more**

sensitive to stratospheric ozone still, the second peak in the upper troposphere has the capability to capture ozone features. AIRS tropospheric ozone retrieval is utilized by various studies to see the events-based ozone enhancements, i.e., Phanikumar et al. (2017) over the balloon launch site (Nainital) utilizes the AIRS ozone measurments to confirm the two folds enhancements of tropospheric ozone due orography induced gravity waves. Li et al. (2018) also utilizes the AIRS middle tropospheric ozone to study the high tropospheric ozone in Lhasa due to convective transport and stratospheric intrusion, etc. Additionally, we studied the ozone variability near 50 hPa (AKs peak altitude) from ozonesonde, which is about 342 ppbv (standard deviation with a mean of about 1630 ppbv), while with logarithmic values, it is 0.2. A typical variability of 20% is seen around the mean ozone mixing ratio at 50hPa.

Phanikumar, D.V., Kumar, K.N., Bhattacharjee, S., Naja, M., Girach, I.A., Nair, P.R. and Kumari, S.: Unusual enhancement in tropospheric and surface ozone due to orography induced gravity waves. *Remote Sensing of Environment*, *199*, pp.256-264, 2017.

Li, D., Vogel, B., Müller, R., Bian, J., Günther, G., Li, Q., Zhang, J., Bai, Z., Vömel, H. and Riese, M.: High tropospheric ozone in Lhasa within the Asian summer monsoon anticyclone in 2013: influence of convective transport and stratospheric intrusions. *Atmospheric chemistry and physics*, *18*(24), pp.17979-17994, 2018.

7) Figure 5: The caption should clarify what do the error bars mean. They should show standard error of the mean not standard deviation. It appears that AIRS provides just the AP value in the troposphere, as one expects from the AKs.

Thanks. Following the reviewer's suggestion, we have now changed the standard deviation by the standard error in the revised Figure 5. In the optimal estimation method, the apriori ozone profiles are modified to match the true atmospheric ozone by minimizing the cost function. Based on the weighting function of particular satellite instruments the apriori is modified at various altitude levels. In general, the ozone weighting function is low in lower troposphere, even the present hyperspectral satellite instruments cannot provide lower tropospheric ozone information. However, there have been some attempts to utilize the synergic observations of infrared and UV-VIS satellites to maximize the retrieval sensitivity to lower tropospheric ozone , as in the case of the synergic ozone retrieval from IASI + GOME-2 (Causta et al., 2013; Rawat et al., 2021) and AIRS + OMI (Fu et al., 2018). In the revised MS (section 3.2) we have briefly described the contribution of apriori in the lower troposphere and the constraint with hyperspectral retrieval.

Cuesta, J., Eremenko, M., Liu, X., Dufour, G., Cai, Z., Höpfner, M., von Clarmann, T., Sellitto, P., Forêt, G., Gaubert, B. and Beekmann, M., 2013. Satellite observation of lowermost tropospheric ozone by multispectral synergism of IASI thermal infrared and GOME-2 ultraviolet measurements over Europe. *Atmospheric Chemistry and Physics*, *13*(19), pp.9675-9693.

**Fu, D., Kulawik, S.S., Miyazaki, K., Bowman, K.W., Worden, J.R., Eldering, A., Livesey, N.J., Teixeira, J., Irion, F.W., Herman, R.L. and Osterman, G.B., 2018. Retrievals of tropospheric ozone profiles from the synergism of AIRS and OMI: methodology and validation.** *Atmospheric Measurement Techniques*, *11*(10), pp.5587-5605.

8) Figure 6: Delete the top panel. (See comment no 4.) The results plotted in the second and 3rd panels are hard to see. To make it clearer remove the error bars (they are not errors anyhow) and the dashed vertical Iines. It is not clear why the data are doubly averaged. If one wants to shows the mean MR in a layer, show just the mean MR without applying AK to the sonde data. This is what a user cares. But if the purpose is to evaluate the AIRS algorithm and calibration, show the MR at a single pressure level after applying AK. (See overall comments.)

**Thank you very much. Following your suggestion, we have now removed the smoothed ozonesonde values, error bars, and vertical lines in figure 6. We agree with the reviewer that the average ozone mixing ratio between the 100-10 hPa region will be dominated by the ozone mixing ratio near the 10hPa region. In the revised MS, we have divided the region into the lower stratosphere (100 - 50 hPa) and middle stratosphere (50 - 10 hPa). In addition, the correlation is now in between the relative difference of ozone mixing ratio (ozonesonde and AIRS) and MI/TWV. Similarly, figure S6 is also revised.**

9) Figure 7: It would be useful to plot the mean difference between sondes and MLS on the left panel. This will tell us if the sondes agree with a much higher vertical resolution satellite instrument. If not this will either imply problems with sonde data or more likely the complexity of doing satellite retrievals near their site. Recommend deleting the middle panel. In the right panel show the std devs desperately from sondes, sonde AK and AIRS to see if AIRS is at least capturing the variability irrespective of the bias. A figure showing r2 would also be useful.

**Thank you for recommending this. The mean biases between ozonesonde and MLS are added in Figure 7, and the middle panel of RMSE is removed in the revised MS. The R2 profile is discussed on the right of figure 8. The mean biases between ozonesonde and MLS, a high vertical resolution satellite instrument, are smaller and MLS agrees well with ozonesonde. We have added the MLS differences with ozonsende in section 3.3. In addition, we find the statistical analysis in the previous MS was by fault, selected for a shorter time. Now, in the revised MS, the complete period from Jan 2011 to Dec 2017 is included in calculating the bias and STD.**

10) Figure 8: Same comment as for Figure 6.

**Thanks. In the revised MS, we have now limited the layer up to 50 hPa, instead of up to 10 hPa. Now it shows the lower stratospheric region (100 - 50 hPa).**

11) Figure 9: same comment as for Figure 7.

**Thanks. We have now removed the middle panel of RMSE and the figure is revised as suggested by the reviewer.**

12) Figure 10: This is arguably the most important figure of the paper. Please try to improve the figure so it is easier to evaluate. See discussion in overall comments.

**Thanks. We have now revised the figure 10. In the revised figure, we have now removed monthly variation with AK as suggested by you in previous comments. We agree that this has also improve its visibility significantly. We have now revised the caption also.**

13) The figure captions should be self-explanatory. One shouldn't be required to hunt in the text to understand the figures.

**Thank you for your suggestions. We have now revised the caption and added the needed information.**

14) The paper requires careful editing. I see citations with no references and references with no citations.

**Thank you for your careful reading and for pointing out the mismatched citation and references. We are sorry for the same and suitable revision has been done in the revised MS.**

---

## Editor Decision (ED1)

**REVIEW OF Rawat et al (2022),** Performance of AIRS ozone retrieval over the central Himalayas: Case studies of biomass burning, downward ozone transport and radiative forcing using long-term observations, FOR AMT

**December 2022**

**SUMMARY & RECOMMENDATION**

This paper uses AIRS (on NASA's Aqua satellite platform) long-term observations of ozone centered over the central Himalayan mountains to: (1) evaluate the AIRS ozone product with ozonesonde and other satellite observations and (2) determine sources for observed trends.  The work is novel as it is a first time analysis of these products in this region using ozonesondes and satellite observations.

The paper does lack a clear, concise final interpretation of the results in the Conclusion section, which makes determining the authors' main conclusions difficult.  Although one specific conclusion mentioned, which is interesting to note, is that lower differences with the ozonesondes are observed in the lower and middle troposphere and stratosphere with nominal underestimations of less than 20%. The abstract does highlight these results, so more attention is needed in the final section of the paper.  Below are listed other several areas where additional information is desired and a few comments. Publication with minor revisions is the recommendation.

**STRENGTHS OF THE PAPER**

There are BLAH important elements of the analysis that make it appealing, original, and thorough:

- The careful statistical analysis of the evaluations between AIRS and the ozonesondes (ground truth) including the use of the satellite averaging kernels to make more accurate comparisons.
- The use of other widely-used satellite products and other satellite IR sensors for comparison to AIRS.
- Histograms that show nicely the vertical and seasonal variability of AIRS as compared to the ozonesondes.
- The attempt to show the application of the AIRS dataset for studying observed trends.

The above analyses and their interpretations explain why this paper merits publication.

**AREAS OF IMPROVEMENT FOR THE REVISED PAPER**

- Generally, there appears to be updates in the references used throughout the paper (latest revised draft), but still found the paper lacking newer publications cited.  For example, there are newer publications for ozonesondes than Smit et al. (2007).  There is the latest ozonesonde report that came out last year (and references therein) that should be cited:  Smit, H. G. J., Thompson, A. M., & the Panel for the Assessment of Standard Operating Procedures for Ozonesondes, v2.0 (ASOPOS 2.0). (2021). Ozonesonde measurement principles and best operational practices, GAW Report 268. World Meteorological Organization. Retrieved from https://library.wmo.int/doc_num.php?explnum_id=10884.
- This leads to the next comment: Has this ozonesonde data been reprocessed to make sure it accounts for an instrument-specific corrections as well as others? Were these ozonesondes

EnSCI or Science Pump or something else?  These corrections need to be applied to the ozonesonde data for comparisons with satellite products to be more accurate (see reference above) and this additional guidebook for data reprocessing: Smit, H. G. J., & the Panel for the Assessment of Standard Operating Procedures for Ozonesondes (ASOPOS). (2012). Guidelines for homogenization of ozonesonde data, SI2N/O3S-DQA activity as part of "Past changes in the vertical distribution of ozone assessment". Retrieved from http://www-das.uwyo.edu/%7Edeshler/NDACC_O3Sondes/O3s_DQA/O3S-DQA-Guidelines%20Homogenization-V2-19November2012.pdf .

- In some of the figures, the ozonesonde data convolved with AIRS averaging kernels looked worse in comparison to AIRS than original ozonesonde data.  No specific comments were noted on this so this should be addressed in the paper.  For example, why would this be case?
- The satellite-derived balloon-burst climatology (McPeters et al., 1997) used to calculate the total ozone column is an outdated climatology.  There is a newer one used more commonly now: McPeters, R. D., & Labow, G. J. (2012). Climatology 2011: An MLS and sonde derived ozone climatology for satellite retrieval algorithms. Journal of Geophysical Research, 117, D10303. https://doi.org/10.1029/2011JD017006.  For example, Stauffer et al (2022) used this in the latest paper on the global ozonesonde network.

    Stauffer, R. M., Thompson, A. M., Kollonige, D. E., Tarasick, D. W., Van Malderen, R., Smit, H. G. J., et al. (2022). An examination of the recent stability of ozonesonde global network data. Earth and Space Science, 9, https://doi.org/10.1029/2022EA002459.

    The use of an older climatology could explain some of the discrepancies observed in total column ozone comparisons and the recommendation is to redo this analysis with more recent climatology.

- Final recommendation: an overhaul is needed on the final section of the paper to state the final conclusions more clearly, similar to what is in the paper abstract.

---

## Author Response (AR2)

**Performance of AIRS ozone retrieval over the central Himalayas: Case studies of biomass burning, downward ozone transport and radiative forcing using long-term observations**

**By Prajjwal Rawat et al., 2022 (AMTD)**

**We would like to thank the referees for their comments and constructive suggestions. The manuscript is suitably revised by incorporating the reviewer's suggestions and comments. Please find here our responses in boldface and the reviewer's comments are in regular font.**

Report #1:

There is no label on the x-axis on Figure 10. My recommendation is to delete information that is widely available from elementary text books, such as the definition of mean, standard deviation, and correlation, or standard physical constants. I have not checked them for accuracy.

**Thank you very much and sorry for missing the label on x-axis. We have now revised Figure 10 and given the months name on the x-axis label. About the information/equations on standard deviation and correlation (mentioned in section 2.3), we have now removed equation 6 (standard deviation) and equation 7 (Pearson's correlation coefficient). However, we have kept the information on RMSE and Bias as they are also providing the information on different RTA layers, calculated for satellite and ozonesonde. Here, we have used weighted statistics that are different than the simple statistics, which will be handy to the readers when comparing gaseous profiles in the atmosphere.**

Report #2:

Authors sincerely replied to my comments, but I still cannot recommend this paper for publication in AMT, due to the insufficient level of the scientific quality.

**Thanks.**

**We do not agree with the comments on the scientific quality. This is for the first time that vertical ozone profiles from satellites have been compared with ozonesonde observations over the Himalayan region. Due to the complex topography of the Himalayas, various surface characteristics change abruptly over the fixed footprint size of the satellites and make it difficult to retrieve atmospheric parameters and composition from satellite instruments. The AIRS averaging kernel is successfully calculated in all the 100 RTA layers using the trapezoid function and utilized for the first time in the evaluation study. It is important to mention that this exercise has been done using 242 ozone soundings for evaluating satellite ozone profiles and identifying the role of downward transport and biomass burning in influencing the ozone profile. At the same time, larger variations in the tropopause height over this subtropical Himalaya were observed during winter and spring when there are frequent tropopause folding events, higher mean biases (~ 65%) and relative difference (~ 150%) between AIRS and ozonesonde in the UTLS region. Further, the AIRS total ozone showed larger differences with ozonesonde for the autumn season and the UTLS column shows persistent biases but higher correlations throughout the months. In summary, such a large period (2011-2017) of ozonesonde data from South Asia are utilized for the first time to make extensive analysis on comparison with satellite data and variations in different height layers. Hence we feel our manuscript has sufficient scientific quality.**

In introduction, this paper emphasized the impatance of evaluating AIRS quality over Himalayas due to its complex topograph. However, I cannot found in section 3 result and discussion where this issue is discussed. They just performed the general comparison, just over the Himalayas.

**We disagree with the reviewer. The terrain pressure and other surface parameters are rapidly varying in the mountain regions of the Himalayas and systematic**

evaluation/validation of satellite measurements are still lacking here. We have emphasized on the importance of evaluating satellite retrieval over the complex terrain regions, explaining all possible factors influencing the satellite ozone retrieval. Recently Cazorla and Herrera (2022) validated various satellite instruments' ozone retrieval with ozonesonde over the Andes Mountains. This study also explained the need of satellite retrieval in the mountain region and studied the possible differences between ozonesonde and satellite measurements. Such evaluation studies along with ours, offer an opportunity to understand the differences between satellite and truth observations over complex terrains and improve satellite retrievals. The evaluation will be equally important for users to find the retrieval credibility of various space-based ozone observations over such regions for their utilization and for the retrieval developer to mitigate such biases. In addition, our results also discussed the influence of downward ozone transport and biomass burning in influencing ozone profile over the subtropical Himalayan region and their identification from satellite and ozonesonde.

More specifically, the AIRS ozone products are analyzed in terms of retrieval sensitivity, retrieval biases/errors, and ability to retrieve the natural variability of columnar ozone, which has not been done so far from the Himalayan region. Section 3.3 discusses the greater difference between low vertical resolution AIRS ozone with ozonesonde in the UTLS region over the Himalayan region, where the various active dynamical process influences ozone, while the high vertical resolution instrument MLS does relatively well. Section 3.4 shows that not only the AIRS retrieval but other IR satellite instruments (AIRS and CrIS) also shows higher differences in the UTLS region. Furthermore, the columnar ozone evaluation is also presented in section 3.5, which shows AIRS total ozone is highly biased compared to ozonesonde and OMI for autumn seasons, while the UTLS and tropospheric column are in reasonable agreement.

*Cazorla, M. and Herrera, E. An ozonesonde evaluation of spaceborne observations in the Andean tropics. Scientific reports, 12(1), pp.1-8, 2022.*

From Figure 4/S4, I still think that comparing AIRS and ozonesonde as a function of horizontal drift is not nessarary (wrong). In troposphere, the horizontal drift is insiginficant and the airmass characteristics are not much different in stratosphere within a few degree. Single ozonesonde profiile should be assumed to be meausred at one location espeically in comparison with satellite measurements.

**We strongly feel that this is a good figure that gives an overview of the region, with a complete information on balloon drift during four seasons. This figure is not for the direct comparison between ozonesonde and AIRS and the same has not mentioned/discussed in this section. Next section (section3.2) describes and compare the vertical distribution in ozone. Additionally, we do not agree that "In troposphere, the horizontal drift is insignificant". This figure clearly shows that balloon drift is different over the Himalayan terrain during four seasons. It moves rapidly in the East, reaching to Nepal, in winter. It slows down in spring and changes direction completely, to go to the West (IGP region of India) in summer.**

**Further, below figure 1 shows the methodology for generating the vertical ozone distribution along the balloon track. It gives both the tropospheric and stratospheric distribution along the balloon track from the ozonesonde and AIRS measurements. Figure S3 and S4 (supplementary) showed the longitudinal/latitudinal variation of biases and $r^2$ among the two measurements, along with the altitude information. We studied bias and $r^2$ variations with coarser (0.5º) and high (0.1º) longitudinal/latitudinal gaps and not much difference in results is observed. This also emphasized the regional distribution of ozone which can be obtained from balloon-borne measurements around the launch site.**

**About the single profile comparisons without horizontal drifts are discussed in sections 3.2 and shown in Figure 5 (in MS), where the ozonesonde and AIRS ozone profiles are compared from surface to 10 hPa.**

[Figure]

**Figure 1. The method to obtain the AIRS ozone along the balloon track.**

Comparison between AIRS and ozonesondes (Figure 6) are linked with monsoon index and total water vaopr, without valid scientific connection. In this figure, I don't think that the monsoon index is not the best parameter for describing this time-series. As stated, the correlation is relatively low (~0.2). I don't understand that the AIRS retrieval quality could be lower, depending on the monsoon index.

**Thank you. We have not claimed any link, on daily or monthly bases, between the monsoon index and AIRS retrieval quality and not described ozone time series based on monsoon index. Rather, this study was to show that when there are weaker monsoon years, the annual average ozone is generally higher compared to the strong monsoon years. We have not discussed or concluded anything on monthly variations or seasonality. We are confirming the finding of Lu et al., 2018 on relation between the tropospheric ozone and monsoon index over the Indian region. To make this further clearer, we have now revised Figure 6 and showed (left lower most corner) annual average values of ozone and monsoon index from year 2011 to 2017.**

I found a lot of cases that the references this paper cited are either "out of date" or "inadequate". It makes me think that the author did not investiate preminary studies well and the credibility of reserach results is low.

**We have now added more references. Below are the clarifications on some confusion on the cited references. We have also revised the sentences accordingly.**

e.g) on page 5, the Himalayas ~~ impacts global/regional radiative budgets and climate pervasively (e.g, Lawrence and Leieveld, 2010; Lelieveld et al. 2018). The cited two papers are about the pollutant outflow from southern Asia and impact of air pollutants on health.

**Thanks, first reference is a kind of review paper for the South Asia that covers all aspects. Nevertheless, we have now added the specific references (Cristofanelli et al., 2014; Zhang et al., 2015) in the revised MS.**

e.g) on page 6, V4 used regression retrieval as the first guess in physical retrieval while later version used a cliamtology based first guess fro the physical retrieval (McPeters et al., 2007).

The cited paper is about an ozone profile cliamtology, not for either the related retrieval algorithm.

**We are sorry for the confusion. The cited reference (McPeters et al., 2007) was not for the retrieval algorithms. This reference was used to refer that the climatology-based first guess is used as a-priori in the satellite retrieval. Now, we have modified this sentence.**

e.g.) Smith et al. 2007 is wrongly cited in this paper for comparing two datasets with different vertical resolutions.

**Thank you for pointing this out. Nevertheless, Smit et al., 2007 was used as this work utilizes ozone data from three different sensors (SPC-5A (EC), SPC-6A (FZJ), and ENSCI-Z (NOAA)) and raw data has different vertical resolutions. We have now added two references (Verstraeten et al., 2013; Boynard et al., 2016) in the section 2.2 that emphasize the need to account vertical sensitivity of two sensors when comparing the ozone profiles.**

In addition, this manuscript should be thoughly revised to correct English usage and many typped errors.

**Thank you. We have revised the MS as much as possible for correct English and for typos.**

---

## Author Response (AR3)

**REVIEW OF Rawat et al (2022), Performance of AIRS ozone retrieval over the central Himalayas: Case studies of biomass burning, downward ozone transport and radiative forcing using long-term observations,**

SUMMARY & RECOMMENDATION

This paper uses AIRS (on NASA's Aqua satellite platform) long-term observations of ozone centered over the central Himalayan mountains to: (1) evaluate the AIRS ozone product with ozonesonde and other satellite observations and (2) determine sources for observed trends. The work is novel as it is a first time analysis of these products in this region using ozonesondes and satellite observations.

The paper does lack a clear, concise final interpretation of the results in the Conclusion section, which makes determining the authors' main conclusions difficult. Although one specific conclusion mentioned, which is interesting to note, is that lower differences with the ozonesondes are observed in the lower and middle troposphere and stratosphere with nominal underestimations of less than 20%. The abstract does highlight these results, so more attention is needed in the final section of the paper. Below are listed other several areas where additional information is desired and a few comments. Publication with minor revisions is the recommendation.

**We thank the reviewer for careful and detailed review of the paper and very constructive feedback. We have revised the manuscript following his/her suggestions, particularly the Conclusion section. Our responses to each suggestion/comment are described below in boldfaces.**

STRENGTHS OF THE PAPER

There are BLAH important elements of the analysis that make it appealing, original, and thorough:
 • The careful statistical analysis of the evaluations between AIRS and the ozonesondes (ground truth) including the use of the satellite averaging kernels to make more accurate comparisons.
 • The use of other widely used satellite products and other satellite IR sensors for comparison to AIRS.
 • Histograms that show nicely the vertical and seasonal variability of AIRS as compared to the ozonesondes.
 • The attempt to show the application of the AIRS dataset for studying observed trends. The above analyses and their interpretations explain why this paper merits publication.

**We appreciate the referee for careful reading and pointing to the strengths of our manuscript.**

AREAS OF IMPROVEMENT FOR THE REVISED PAPER
• Generally, there appears to be updates in the references used throughout the paper (latest revised draft), but still found the paper lacking newer publications cited. For example, there are newer publications for ozonesondes than Smit et al. (2007). There is the latest ozonesonde report that came out last year (and references therein) that should be cited: Smit, H. G. J., Thompson, A. M., & the Panel for the Assessment

of Standard Operating Procedures for Ozonesondes, v2.0 (ASOPOS 2.0). (2021). Ozonesonde measurement principles and best operational practices, GAW Report 268. World Meteorological Organization. Retrieved from https://library.wmo.int/doc_num.php?explnum_id=10884.

**Thank you very much. We have added these new references in the revised manuscript.**

• This leads to the next comment: Has this ozonesonde data been reprocessed to make sure it accounts for an instrument-specific corrections as well as others? Were these ozonesondes EnSCI or Science Pump or something else? These corrections need to be applied to the ozonesonde data for comparisons with satellite products to be more accurate (see reference above) and this additional guidebook for data reprocessing: Smit, H. G. J., & the Panel for the Assessment of Standard Operating Procedures for Ozonesondes (ASOPOS). (2012). Guidelines for homogenization of ozonesonde data, SI2N/O3S-DQA activity as part of "Past changes in the vertical distribution of ozone assessment". Retrieved from http://wwwdas.uwyo.edu/%7Edeshler/NDACC_O3Sondes/O3s_DQA/O3S-DQAGuidelines%20Homogenization-V2-19November2012.pdf .

**Thanks. We are using EN-SCI ozonesonde, which was developed by Dr. Komhyr (Komhyr, 1969; Komhyr et al. 1995), and this sensor is being used extensively worldwide. Briefly, EN-SCI ECC ozonesonde coupled with iMet radiosondes are operated under the standard operating procedures (SOPs) as documented by EN-SCI documentation and described elsewhere. The ASOPOS recommended sensing solution type for ENSCI with 0.5% KI half buffer (SST0.5) is used. The ozonesonde sensor's successful performance is assured before launch (about 3 - 7 days before launch) as part of advance preparation and during the day of launch by maintaining and reviewing the records for background current, pump flow rate, response time, etc. Here we have also checked the background current variation during 2011 - 2017, and it was below 0.08 μA with an average of 0.025 ± 0.012 μA, which is as suggested (Smit & ASOPOS Panel, 2020; Ancellet et al., 2022) for acceptable launch. Furthermore, the total ozone normalization factor (NF) is exclusively used to screen the overall quality of the ozonesonde ozone profile. These details are already made available in earlier publications (Rawat et al., 2020) and we missed to mention them here. Now we have added this information briefly in the revised MS (section 2.14).**

**According to ASOPOS recommendation, if NF lies in the following range 0.9<NF<1.2, the data is considered of good quality (Smit & ASOPOS Panel, 2020). We have now calculated the total ozone normalization factor for our ECC ozonesonde with collocated OMI total ozone as reference. The total ozone from ECC ozonesonde is estimated by integrating ozone up to burst altitude and then using a balloon burst climatology from McPeters and Labow (2012). Figure 1 below shows the variation of the total ozone normalization factor for our ozonesonde from 2011 to 2017 and the respective frequency distribution on the right side.**

This factor is well within the ASOPOS recommendation with an average of **1.0 ± 0.04**, which implies the reasonable quality of our ozonesonde. The use of such correction factors has been in debate for a long time as ECC ozonesonde are used as the independent measurements of ozone (Smit & ASOPOS Panel, 2020), however used for quality checks. Additionally, ozonesonde observations from present site have also been utilized in various campaigns based studies [SUSKAT (Bhardwaj et al., 2018), StratoClim (Brunamonti et al., 2018)] and in other studies (Ojha et al., 2014; Ojha et al., 2017). We have now added this information in the revised manuscript (section 2.1.4).

Ancellet, G., Godin-Beekmann, S., Smit, H.G., Stauffer, R.M., Van Malderen, R., Bodichon, R. and Pazmino, A. Homogenization of the Observatoire de Haute Provence electrochemical concentration cell (ECC) ozonesonde data record: comparison with lidar and satellite observations. *Atmospheric Measurement Techniques*, *15*(10), pp.3105-3120, 2022.

Ojha, N., Pozzer, A., Akritidis, D., and Lelieveld, J.: Secondary ozone peaks in the troposphere over the Himalayas, Atmos. Chem. Phys., 17, 6743–6757, https://doi.org/10.5194/acp-17-6743-2017, 2017.

**(Here, references do not present in the manuscript are provided)**

[Figure]

**Figure 1. Estimated total ozone normalization factor of ECC ozonesonde with the Aura OMI satellite instrument. Corresponding frequency histograms are also shown on the right.**

• In some of the figures, the ozonesonde data convolved with AIRS averaging kernels looked worse in comparison to AIRS than original ozonesonde data. No specific comments were noted on this so this should be addressed in the paper. For example, why would this be case?

**Thank you for pointing this. Generally, for a perfect instrument and accurate retrieval algorithm, the a-priori contribution in final retrieval is assumed to be minimal, while in poor retrieval, the AKs tend to be zero metrics. Therefore, when the satellite retrieval is poor, or satellite AKs tend to be zero, it can be seen from Eq. 1 that the convolved ozonesonde will be**

**weighted more towards the a-priori. In such cases, the ozonesonde data convolved with AIRS averaging kernels will be different than the original ozonesonde data and may not compare well with AIRS. In the revised manuscript, in section 2.2, we have mentioned such possible profile changes of ozonesonde after applying AKs convolution.**

• The satellite-derived balloon-burst climatology (McPeters et al., 1997) used to calculate the total ozone column is an outdated climatology. There is a newer one used more commonly now: McPeters, R. D., & Labow, G. J. (2012). Climatology 2011: An MLS and sonde derived ozone climatology for satellite retrieval algorithms. Journal of Geophysical Research, 117, D10303. https://doi.org/10.1029/2011JD017006. For example, Stauffer et al (2022) used this in the latest paper on the global ozonesonde network. Stauffer, R. M., Thompson, A. M., Kollonige, D. E., Tarasick, D. W., Van Malderen, R., Smit, H. G. J., et al. (2022). An examination of the recent stability of ozonesonde global network data. Earth and Space Science, 9, https://doi.org/10.1029/2022EA002459.

The use of an older climatology could explain some of the discrepancies observed in total column ozone comparisons and the recommendation is to redo this analysis with more recent climatology.

**Thank you for pointing this. We are sorry for mentioning older reference. We have used the latest data of balloon burst climatology from McPeters and Labow (2012) retrieved from https://acd-ext.gsfc.nasa.gov/anonftp/toms/ML_climatology/, but by mistake we have provided the older references of 1997. We have now revised the references.**

• Final recommendation: an overhaul is needed on the final section of the paper to state the final conclusions more clearly, similar to what is in the paper abstract.

**Thank you for your suggestions. We have revised the conclusion section in the revised manuscript. Particularly in the revised conclusion (1) We have added the improvement in biases in percentage after applying AIRS averaging kernel information to ozonesonde during summer monsoon season (2) We have specifically mentioned the lower weighted statistical (less than 20 %) error in lower-middle troposphere and stratosphere between AIRS retrieved ozone profiles and ozonesonde compared to upper troposphere. (3) The ozone histogram differences between AIRS and Ozonesonde (AK) is added. (4) We also mentioned the higher biases in the upper troposphere for IASI and CrIS ozone retrieval. (5) AIRS capability to capture ozone enhancements of 5 -20% after biomass burning and downward transport events is explained. (6) Lastly, we have also emphasized the study's importance and improved the conclusion's last paragraph in the revised manuscript.**

---

## Author Response (AR4)

**MS# AMT-2022-187**, **Performance of AIRS ozone retrieval over the central Himalayas: Case studies of biomass burning, downward ozone transport and radiative forcing using long-term observations, Rawat et al.,**

- The title should be corrected, particularly for the phrase after colon. I cannot see the significant analyses about the 'Case studies of biomass burning, downward ozone transport and radiative forcing using long-term observations' in this manuscript. Only 2-3 short chapters are related to the case studies, but I cannot see the meaningful finding about the ozone pattern associated with the biomass burning. There is not many obvious explanations of ozone variation related to the biomass burning pattern. In particular, there is not any statement about the biomass burning in the short summary and abstract, showing that authors did not pay attention to this topic. Most of materials are just about the 'Validation of AIRS ozone retrieval over the central Himalayas using ozonesonde and other satellite dataset'. This validation does not look bad so it can be acceptable as a journal paper. Thus, authors should change the title considering the main point of this manuscript.

**Thank you very much for your time in reviewing the MS. As suggested, we have now changed the title of manuscript to "Performance of AIRS ozone retrieval over the central Himalayas: Use of ozonesonde and other satellite dataset".**

- Similar to the first comment, it is very hard to see the meaning of 'Himalayan research'. The only point that authors addressed is that this is the first work over this region. It does not provide a motivation of this study. Authors should suggest the 'unique point' of this work different from other AIRS ozone validation, and this unique point should be excavated using the keyword of 'Himalaya'. Please improve the introduction and conclusion part in this context.

**Thank you very much. We have now added more relevant sentences to further improve the introduction and conclusion part. We feel that now the importance of the region and motivation of the present study is clearer.**

- The manuscript looks unnecessarily long. Please organize whole manuscript again to have a proper length. Authors may need to move some trivial things to the supplement materials (Title change is also related to this length control).

**Thanks. We have now moved most part of the methodology section to the supplement materials. We have also moved one figure to the supplementary section, one figure is modified to reduce its size and few references are reduced.**

- It is better to include Fig. 1 in the author's response (associated with the normalization factor) in the supplement file.

**Thank you for your suggestion. We have now included the figure in the supplement materials as Figure S3.**